# A Meta-Learning Approach to Bayesian Causal Discovery

**Anish Dhir**
Imperial College London
anish.dhir13@imperial.ac.uk

**Matthew Ashman**
University of Cambridge

**James Requeima**
University of Toronto

**Mark van der Wilk**
University of Oxford

## Abstract

Discovering a unique causal structure is difficult due to both inherent identifiability issues, and the consequences of finite data. As such, uncertainty over causal structures, such as those obtained from a Bayesian posterior, are often necessary for downstream tasks. Finding an accurate approximation to this posterior is challenging, due to the large number of possible causal graphs, as well as the difficulty in the subproblem of finding posteriors over the functional relationships of the causal edges. Recent works have used Bayesian meta learning to view the problem of posterior estimation as a supervised learning task. Yet, these methods are limited as they cannot reliably sample from the posterior over causal structures and fail to encode key properties of the posterior, such as correlation between edges and permutation equivariance with respect to nodes. To address these limitations, we propose a Bayesian meta learning model that allows for sampling causal structures from the posterior and encodes these key properties. We compare our meta-Bayesian causal discovery against existing Bayesian causal discovery methods, demonstrating the advantages of directly learning a posterior over causal structure.

## 1 Introduction

Learning causal structure has the potential to supercharge areas of science with the ability to understand the impact of interacting with complex systems. Learning causal structures with certainty, requires interventional data (Pearl, 2009), which is often impossible or very costly to gather. This makes it preferable to infer as much of the causal structure as possible using observational data, and find a way of dealing with the resulting multiple solutions that cannot be distinguished. One way to do this is to use Bayesian inference to express uncertainty over causal structure, which can then be incorporated in downstream tasks, such as effect estimation (Toth et al., 2022), or to increase confidence about relations by collecting more data (Agrawal et al., 2019; Zhang et al., 2023; Jain et al., 2023; Tigas et al., 2023).

Dhir et al. (2024) showed that the Bayesian posterior can distinguish causal structure, even within a Markov equivalence class, using only the implications of priors on functional relationships between variables. This effect occurs because the factorisation assumptions in the prior over functions, implied by the Independent Causal Mechanism (ICM) principle (Janzing & Schölkopf, 2010), can contribute to the inference over the causal structure. Hence, inaccuracies in functional inference can also result in inaccurate inference over causal structure. Current methods that approximate the posterior on causal graphs make restrictive assumptions on the functions for accurate inference (Annadani et al., 2021; Cundy et al., 2021), or rely on approximating the posterior over functions in Bayesian neural networks (Lorch et al., 2021; Annadani et al., 2024), which is notoriously difficult to do accurately (Wenzel et al., 2020). Moreover, the high dimensionality of graphs with many variables further complicates effective inference.

In contrast, Bayesian meta-learning models such as neural processes (Garnelo et al., 2018) directly learn a map from a dataset to a posterior distribution. The Bayesian prior is implicitly encoded

through a distribution over datasets, while directly predicting the posterior implicitly marginalises out any latent variables (Requeima, 2023; Müller et al., 2021). This approach circumvents the need to approximate distributions on functions that is present in previous approaches. Such generative models have also shown strong performance in sampling from high-dimensional distributions, making them well-suited for large graphs. However, directly learning a posterior over causal graphs does introduce new challenges from a generative modelling perspective. We need to allow for complex distributions with dependencies between edges, while also constraining to the space of acyclic graphs (Zheng et al., 2018). Additionally, properties present in the true posterior, such as permutation equivariance with respect to the nodes, must be incorporated. Recent efforts to use neural processes for this purpose only targeted the maximum a posteriori value, leading to limitations when considering the full posterior. Some methods lack the ability to explicitly generate acyclic samples while ignoring permutation equivariance (Lorch et al., 2022), while others fail to capture the dependencies between edges (Ke et al., 2022), resulting in an inability to sample from the correct posterior. Thus, their usefulness for downstream tasks, where the full posterior over causal structures is required, is limited.

We propose the *Bayesian Causal Neural Process* (BCNP) model that addresses these drawbacks. This takes the form of an encoder-decoder transformer that encodes dependencies between edges, and is permutation equivariant with respect to the nodes. Building on recent advances in Charpentier et al. (2021) and Annadani et al. (2024), our decoder allows for direct sampling over directed acyclic graphs (DAGs), by parametrising a distribution over permutation and lower triangular Bernoulli matrices. The samples are thus acyclic by construction and can allow for sampling multiple valid causal structures that are consistent with a given dataset. We show that the BCNP model generates accurate posterior samples compared to previous Bayesian meta-learning approaches (section 4.1). We also demonstrate that the BCNP model outperforms explicit Bayesian models, as well as other meta-learning models when the model is trained on the correct data distribution (section 4.2), and also when the data distribution of a dataset is unknown (section 4.3). Our contributions are as follows: 1) We build on previous insights on meta-learning for causal discovery but focus on estimating the full posterior over causal structures. We introduce a decoder and a loss function to achieve this all the while encoding key properties of the posterior (table 1). In contrast with other approaches, this allows for sampling from the correct posterior over causal structures. 2) We show that our changes lead to significant performance increases as well as outperforming explicit Bayesian causal models as well as other meta-learning models on synthetic (section 4.2) and semi-synthetic data (section 4.3). We provide code for our method: `CausalStructureNeuralProcess`.

## 2 BACKGROUND AND RELATED WORK

### 2.1 LEARNING CAUSAL STRUCTURE

We denote a given a dataset of $N$ samples with $D$ variables as $\mathbf{X}$. We assume that the dataset is generated according to a Structural Causal Model (SCM), which is specified by a *directed acyclic graph* (DAG) encoded through its acyclic adjacency matrix $\mathcal{G} \in \{0,1\}^{D \times D}$. The variables are generated by following the DAG as

$$X_d = f_d(X_{\mathrm{PA}_\mathcal{G}(d)}, \epsilon_d), \ \text{ for } \ d \in \{d, \dots, N\} \tag{1}$$

where $X_d$ is the $d^{\text{th}}$ variable, $f_d$ is a function, $\mathrm{PA}_\mathcal{G}(d)$ is the index set of parents of variable $X_d$ in the DAG $\mathcal{G}$, and $\epsilon_d$ is some arbitrary noise. Our goal is to infer the DAG (or causal structure) that generated a given dataset.

To do this, we specify a *Bayesian causal model* $P(\mathbf{X}, \mathbf{f}, \mathcal{G}) = P(\mathbf{X}|\mathbf{f}, \mathcal{G})P(\mathbf{f}, \mathcal{G})$ that specifies our belief of the data generation process, with likelihood $P(\mathbf{X}|\mathbf{f}, \mathcal{G})$, and priors over functions $P(\mathbf{f})$ and over DAGs $P(\mathcal{G})$. In the SCM, changing a variable's functional mechanism does not change any other variable's functional mechanisms. This imposes a constraint on the Bayesian causal model in that it must satisfy the *Independent Causal Mechanism* (ICM) assumption (Janzing & Schölkopf, 2010). This is achieved by ensuring that prior over functions factorise over the variables, $P(\mathbf{f}) = \prod_{d=1}^{D} P(f_d)$ (Stegle et al., 2010; Dhir et al., 2024). The posterior, which specifies the model's belief over DAGs can be computed as

$$P(\mathcal{G}|\mathbf{X}) = \frac{P(\mathbf{X}|\mathcal{G})P(\mathcal{G})}{\sum_\mathcal{G} P(\mathbf{X}|\mathcal{G})P(\mathcal{G})}, \tag{2}$$

where $P(\mathbf{X}|\mathcal{G})$ is referred to as the *marginal likelihood*. Crucially, the marginal likelihood is the normalization factor for the posterior distribution over the model's functional mechanisms $\mathbf{f}$ and thus relies on accurate posterior inference over $\mathbf{f}$. A Bayesian causal model is called *identifiable* if a dataset it generates corresponds uniquely to the correct DAG (Guyon et al., 2019, Ch. 2). In such cases, Dhir et al. (2024) demonstrated that, under the ICM assumption, the marginal likelihood—and by extension, the posterior over functions—is sufficient to distinguish causal directions. Thus, inaccurate functional inference can lead to incorrect causal structures being inferred. Furthermore, if a causal model is not identifiable, multiple causal structures may explain the dataset to varying degrees — a challenge that also arises with finite samples. In these cases, the posterior over causal structures in eq. (2) quantifies this uncertainty.

Although the marginal likelihood is tractable for simpler models, this can introduce misspecification issues. For more flexible function approximators, such as neural networks, computing the marginal likelihood becomes challenging due to the complexity of functional inference. Additionally, the large space of possible DAGs further complicates this process, as the normalizer in eq. (2) is intractable to compute, and sampling efficiently in high-dimensional spaces is difficult.

## 2.2 RELATED WORK

A lot of causal discovery focuses on identifying a single causal structure that generated a dataset (Lachapelle et al., 2019; Rolland et al., 2022; Zhang et al., 2015). This is prone to errors in cases where, due to identifiability or finite sample issues, multiple causal structures can explain a dataset. On the other hand, Bayesian causal discovery provides a distribution over possible causal structures that align with the observed data. As our work is concerned with the latter, we review existing approaches in this area.

The challenges of Bayesian causal discovery lie in accurate estimation of the marginal likelihood, which requires accurate inference over functional mechanisms and the large number of graphs. Geiger & Heckerman (2002) addressed this by limiting their approach to linear Gaussian models and constraining to a single prior, ensuring identical posteriors within a Markov equivalence class, thereby reducing the search space. Other methods for linear models enable the incorporation of additional priors by using variational inference to tackle the large search space (Cundy et al., 2021; Annadani et al., 2021). However, even though inference is accurate, considering only simple functional relationships may lead to misspecification. For more expressive models, such as neural networks, methods that combine both sampling and variational inference have been proposed. DiBS (Lorch et al., 2021) uses Bayesian neural networks to model relationships between variables and Stein variational gradient descent (Liu & Wang, 2016) to perform inference over both the neural networks and the causal structure. A regulariser is used to ensure that the samples are acyclic (Zheng et al., 2018). However, this approach is computationally expensive with quadratic cost with the number of samples and further samples requires restarting the optimization process. In contrast BayesDAG (Annadani et al., 2024) directly parametrise the space of DAGs using upper triangular and permutation matrices (Charpentier et al., 2021), eliminating the need for a regulariser and enabling the use of variational inference. For expressivity, they also use Bayesian neural networks, but use SG-MCMC (Chen et al., 2014) to perform inference over the network weights. Despite these advancements, the accuracy of inference mechanisms for Bayesian neural networks remains uncertain (Wenzel et al., 2020), and as such the quality of the posterior over graphs. Our approach differs by amortizing the entire inference process, which mitigates inaccuracies in the inference of functional relationships between variables and facilitates rapid sampling from the posterior distribution.

The line of work closest to ours is Bayesian meta-learning based causal discovery, where causal discovery is reformulated as a classification problem. Lopez-Paz et al. (2015) first proposed this with two variables. The kernel mean embedding of data with known causal relationships (or synthetically generated) was used to train a classifier. Li et al. (2020) extended the approach using neural network embeddings to output the adjacency matrix for multiple variables. A key observation was that the adjacency matrix must be permutation equivariant with respect to the variables, improving the model's statistical efficiency. However, they limited their input features to the correlation coefficients between variables. Recent advances like AVICI (Lorch et al., 2022) and CSIvA (Ke et al., 2022) extend the above approach, using transformers on learnt embeddings of the inputs instead of fixed features. Here, attention between nodes and samples of a node are interleaved, ensuring that the final representation is extracted such that it is permutation invariant with respect to the samples.

AVICI uses a max pooling operation to achieve this, while CSIvA uses an attention operation (Lee et al., 2019). They both also differ in their representation of the adjacency matrix. AVICI decodes the representation using a linear layer to a binary matrix, while CSIvA uses an autoregressive decoder to estimate the probability of each element in sequence.

Methods such as AVICI and CSIvA aim to approximate the posterior over the causal structure (Ke et al., 2022; Lorch et al., 2022), but only target the maximum a posteriori value. Thus, when considering the full posterior, these approximations fail to capture essential properties, resulting in inaccurate estimates. AVICI only estimates the marginal probabilities of an edge existing, while the use of an acyclicity regulariser directly on the probabilities biases these estimates. Moreover, AVICI can also only provide acyclic samples in expectation and does not provide samples from the actual posterior (section 4.1). CSIvA captures dependencies between edges but lacks permutation equivariance, meaning that reordering nodes can change its belief about causal relationships It also does not guarantee acyclic samples, and its autoregressive approach performs poorly as the number of variables increases. In this work, we allow for sampling acyclic graphs while retaining the performance of these methods. The differences between AVICI, CSIvA, and ours (BCNP) can be seen in table 1.

Table 1: Comparison of our approach against existing Bayesian meta-learning methods for Bayesian causal discovery (AVICI and CSIvA).

|  | AVICI | CSIvA | BCNP (ours) |
|---|---|---|---|
| Acyclic Samples | In expectation | No | Yes |
| Edge Dependency | No | Yes | Yes |
| Permutation Equivariance | Yes | No | Yes |

## 3  CAUSAL STRUCTURE LEARNING AS BAYESIAN META LEARNING

The process described in eq. (2) requires positing a causal model including prior beliefs over the causal functions, as well as the causal structure. Due to the presence of the marginal likelihood term in eq. (2), the posterior over the causal structure also requires computing the posterior over the functional parameters. For flexible models, such as neural networks, good approximations to this posterior are hard to obtain. To address this difficulty, we can use neural processes (Garnelo et al., 2018) to learn a direct mapping from datasets to posteriors over causal graphs, bypassing the need to approximate the posterior over functions.

Our goal is to approximate the posterior of the Bayesian causal model of choice, which we refer to as $P_{\text{BCM}}(\mathbf{X}, \mathbf{f}, \mathcal{G})$ — described in section 2.1. To do so, we use a *Bayesian Causal Neural Process* (BCNP) model $P_\phi(\mathcal{G}|\mathbf{X})$ with parameters $\phi$. This model is an encoder-decoder transformer network that takes in a dataset and directly approximates the posterior $P_{\text{BCM}}(\mathcal{G}|\mathbf{X})$. To train it to do so, we minimise the KL-divergence between $P_{\text{BCM}}(\mathcal{G}|\mathbf{X})$ and $P_\phi(\mathcal{G}|\mathbf{X})$,

$$\min_\phi -\mathbb{E}_{P_{\text{BCM}}(\mathbf{X})}\left[\text{KL}\left[P_{\text{BCM}}(\mathcal{G}|\mathbf{X})\|P_\phi(\mathcal{G}|\mathbf{X})\right]\right] \tag{3}$$

$$=\min_\phi -\mathbb{E}_{P_{\text{BCM}}(\mathcal{G})}\left[\mathbb{E}_{P_{\text{BCM}}(\mathbf{f})}\left[\mathbb{E}_{P_{\text{BCM}}(\mathbf{X}|\mathbf{f},\mathcal{G})}\left[\log P_\phi(\mathcal{G}|\mathbf{X})\right]\right]\right] + C \tag{4}$$

where $C$ is a constant that does not depend on $\phi$. During training, the expectation in eq. (3) is computed via a Monte Carlo approximation that requires sampling from the Bayesian causal model $P_{\text{BCM}}(\mathbf{X}, \mathbf{f}, \mathcal{G})$. Sampling from this model proceeds by first sampling a graph $\mathcal{G} \sim P_{\text{BCM}}(\mathcal{G})$, then a functional mechanism for each of the $D$ variables $\mathbf{f} = \{f_1, \ldots, f_D\} \sim P_{\text{BCM}}(\mathbf{f})$, and then $N$ samples of each variable generated according to $\mathcal{G}$ denoted $\mathbf{X} \sim P_{\text{BCM}}(\mathbf{X}|\mathbf{f}, \mathcal{G})$. The above procedure applies not only to datasets and causal graphs from an explicitly defined Bayesian causal model but also enables posterior approximation from datasets and causal graphs generated from some unknown distribution. Regardless of the data generating procedure, by directly learning the posterior over graphs, the model $P_\phi(\mathcal{G}|\mathbf{X})$ implicitly marginalises out the function $\mathbf{f}$, removing the need to approximate the posterior over functions. The above KL-divergence is minimised if and only if the model recovers the true posterior, $P_\phi(\mathcal{G}|\mathbf{X}) = P_{\text{BCM}}(\mathcal{G}|\mathbf{X})$ (Foong et al., 2020).

There are certain properties of the posterior $P_{\text{BCM}}(\mathcal{G}|\mathbf{X})$ that can be encoded directly in $P_\phi(\mathcal{G}|\mathbf{X})$. Permuting the samples of a dataset does not change the posterior distribution, which implies that $P_\phi(\mathcal{G}|\mathbf{X})$ must be permutation-invariant with respect to the samples. Furthermore, permuting the nodes in the dataset should not change the model's belief, expressed through the posterior, about the existence of a causal relationship. For example, if a model predicts $X \rightarrow Y$ with a certain probability when given $(X, Y)$ as inputs, it should predict the same relation with the same probability if the inputs are permuted to $(Y, X)$. As causal relationships are expressed through a DAG $\mathcal{G} \in \{0, 1\}^{D \times D}$, this requires the probability of sampling a DAG to be permutation-equivariant with respect to the nodes.

In addition to encoding these properties, the model must also be able to sample DAGs from the posterior to be practically useful. Next, we introduce a network that parameterises the distribution $P_\phi(\mathcal{G}|\mathbf{X})$ to represent a distribution over DAGs, allowing us to sample acyclic graphs corresponding to valid causal structures. Furthermore, it also ensures that the distribution is permutation invariant with respect to the samples, and permutation equivariant with respect to the nodes.

## 3.1 ENCODER

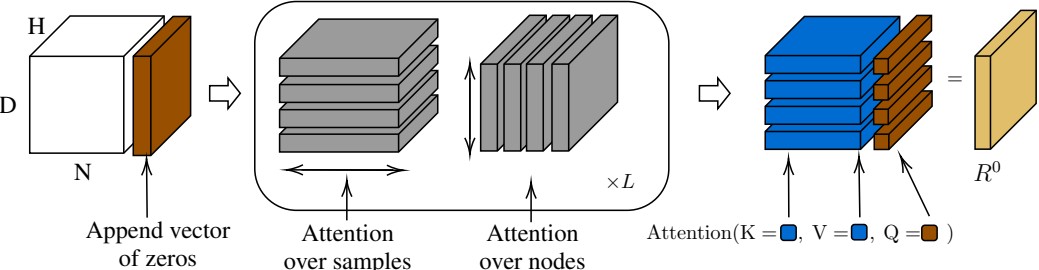

Figure 1: Each dataset contains $D$ nodes and $N$ samples where each data point is embedded into a vector of size $H$, giving a $D \times N \times H$ tensor. A *query vector* of zeros is then appended along the sample axis. The data is passed through $L$ transformer layers which alternate between attention over samples and attention over nodes. The summary representation $R^0$ is constructed using an attention layer where the samples of each node serve as the keys and values and the query vector acting as the query.

Our encoder follows that of Ke et al. (2022); Lorch et al. (2022). Inputs are embedded and passed onto transformer layers (Vaswani et al. (2017), see appendix A.1) that alternate between attention across samples and attention across nodes. Finally a summary representation vector is constructed that summarises the information in each node across all the samples. Crucially, the summary representation vector is constructed such that it is permutation invariant with respect to the samples of each node. Figure 1 visualises the encoder.

**Embedding:** Each input node and sample $x_{i,d}$ is embedded into a vector of size $H$ using a linear map. As we want the subsequent attention operations to summarise the information across the samples, we concatenate a vector of 0s, which we call the *query vector*, in the sample dimension. This gives a tensor of size $\mathbf{H}^0 \in \mathbb{R}^{D \times N+1 \times H}$.

**Transformer layers:** The tensor $\mathbf{H}^0$ is then passed onto a transformer layer that performers attention across samples. That is, parameters are shared across nodes. Following this, a transformer layer is applied that performs attention across nodes, with parameters across samples being shared. This procedure is repeated $L$ times.

**Summary representation:** The representation of the data is extracted from the output of the transformer layers $\mathbf{H}^L \in \mathbb{R}^{D \times N \times H}$, along with the query vector $\mathbf{q} \in \mathbb{R}^{D \times 1 \times H}$. A final cross-attention operation across samples is carried out using $\mathbf{H}^L$ as the key and values, and $\mathbf{q}$ as the query. The output of this is a summary representation vector $\mathbf{R}^0 \in \mathbb{R}^{D \times H}$ that summarises the information in the samples for each node.

The cross-attention operation ensures that the summary representation $\mathbf{R}^0$ is permutation invariant with respect to the samples (Kim et al.). As the nodes are only processed using self-attention, the summary representation is also permutation equivariant with respect to the nodes (Lee et al., 2019).

## 3.2 DECODER

One of the main differences in our architecture from those in Ke et al. (2022); Lorch et al. (2022) lies in the design of our decoder. Unlike the decoders in these prior works, our proposed decoder is designed to directly sample from the distribution of interest, requiring us to parameterize a distribution over DAGs. Charpentier et al. (2021) show that any DAG $\mathcal{G}$ can be represented by a permutation matrix $\mathbf{Q}$, and a lower triangular binary matrix $\mathbf{A}$ through $\mathcal{G} = \mathbf{Q}\mathbf{A}\mathbf{Q}^T$. This ensures that samples are acyclic and hence represent valid causal structures. Thus, we can reduce the problem of learning a distribution over DAGs to one of learning a distribution over permutations and over lower triangular binary matrices.

**Distribution over Permutations:** Permutation matrices, $\mathbf{Q}$, can be parametrised by a matrix $\Theta \in \mathbb{R}^{D \times D}$ by solving the following non-differentiable problem

$$\mathbf{Q} = \arg\max_{\mathbf{Q}'} \langle \mathbf{Q}', \Theta \rangle, \tag{5}$$

where $\langle A, B \rangle = \text{trace}(A^T B)$ is the Frobenius inner product of matrices. Intuitively, eq. (5) yields the permutation matrix, which when applied to $\Theta$, maximises the trace. Changing the values of $\Theta$ then changes the permutation matrix that yields the maximum. To allow for learning with gradients, Mena et al. (2018) propose modifying the problem in eq. (5) using the differentiable *Sinkhorn* operator $\text{S}(\cdot)$. The Sinkhorn operator consists of repeated normalisations of the rows and columns of the input, converging to a solution of eq. (5) in the limit. Mena et al. (2018) show that the parametrisation of the permutation matrix in eq. (5) can be well approximated by $\text{S}\left(\frac{\Theta}{\tau}\right)$ for $\tau \to 0$, where $\tau$ is some temperature parameter. Further, adding Gumbel noise $\mathbf{G}$ gives a differentiable distribution over permutation matrices called the *Gumbel-Sinkhorn*, $\mathcal{GS} := \text{S}\left(\frac{\Theta}{\tau} + \mathbf{G}\right)$. This distribution does not have a tractable density but allows for sampling permutations parametrised by $\Theta$. We follow Annadani et al. (2024) and parametrise $\Theta$ using a low rank representation that takes fewer sinkhorn iterations to converge to a solution. The representation from the encoder $\mathbf{R}^0 \in \mathbb{R}^{D \times H}$ is first processed using transformer layers to give $\mathbf{R}^{L_1}$, and then $\Theta$ is parametrised as (Annadani et al., 2024)

$$\Theta = \mathbf{R}^{L_1} \mathbf{o}^T, \quad \mathbf{Q} \sim \mathcal{GS}(\Theta) \tag{6}$$

where $\mathbf{o} = [1, \ldots, D]$. As the rows of $\mathbf{R}^{L_1}$ are permutation equivariant with respect to the nodes, so are the rows of $\Theta$. In practice we use $\tau > 0$ and use the Hungarian algorithm to get discrete permutation matrices for the forward pass (Charpentier et al., 2021). In the backward pass, we use the straight through estimator to optimise through $\mathbf{R}^{L_1}$ (Bengio et al., 2013).

**Distribution over Lower Triangular Matrix:** We model the presence of edges as a lower triangular matrix of Bernoulli random variables. Permuting this lower triangular matrix then orients the edges all the while ensuring acyclicity. To ensure permutation equivariance of this matrix with respect to the nodes, we borrow ideas from attention. Further, to ensure dependence between the permutation and lower triangular binary matrices, we ensure that the distributions share parameters. This is done by processing $\mathbf{R}^{L_1}$ by further transformer layers to give $\mathbf{R}^{L_2}$, which is then used to parametrise the lower triangular matrices through the following operations

$$\text{ParameterAttention}(\mathbf{Q}, \mathbf{K}) := \mathbf{Q}\mathbf{K}^T / \sqrt{H} \in \mathbb{R}^{D \times D}, \tag{7}$$

$$\text{head}_m(\mathbf{R}^{L_2}) := \text{ParameterAttention}(\mathbf{R}^{L_2}\mathbf{W}_m^q, \mathbf{R}^{L_2}\mathbf{W}_m^k), \tag{8}$$

$$\text{MHPA}(\mathbf{R}^{L_2}) := \text{stack}_{m=1,\ldots,M}(\text{head}_m(\mathbf{R}^{L_2}))\mathbf{W}_o, \tag{9}$$

where $\mathbf{W}_o \in \mathbb{R}^{M \times 1}$, $\mathbf{W}_h^q \in \mathbb{R}^{H \times h_k}$, and $\mathbf{W}_m^k \in \mathbb{R}^{H \times h_k}$. ParameterAttention takes in two matrices and computes the outer product giving a $D \times D$ matrix. MHPA is a multi-head variant of ParameterAttention. The matrix outputted by ParameterAttention and thus also by MHPA is permutation equivariant with respect to the input rows (Lee et al., 2019). As we input a linear map of $\mathbf{R}^{L_2}$ into these operations, the output is permutation equivariant with respect to the nodes. The

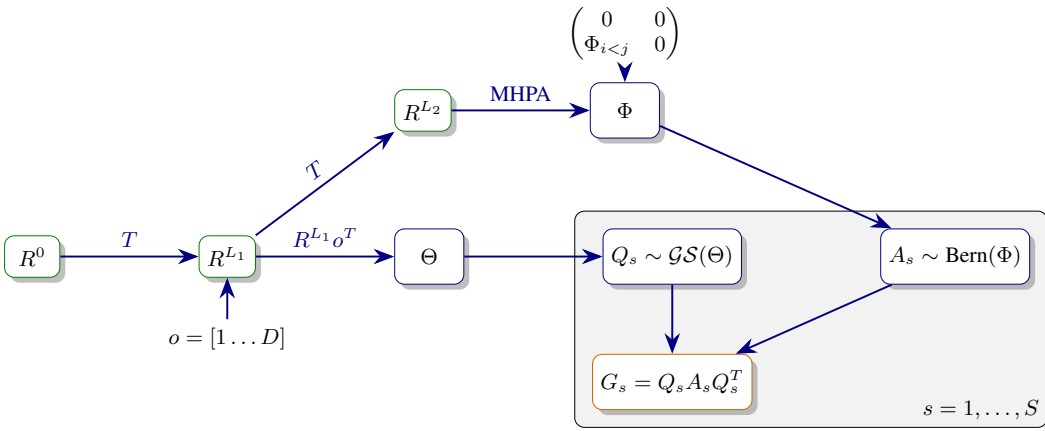

Figure 2: Computational graph of the decoder described in section 3.2. The decoder takes in the summary representation from the encoder $R^0$ as input. $T$ denotes a transformer layer, MHPA denote multi headed parameter attention (eq. (9)), and $\mathcal{GS}$ is the Gumbel-Sinkhorn distribution (Mena et al., 2018). The network outputs samples of permutation matrices $Q_s$ and lower triangular binary matrices $A_s$ that can be used to construct samples of DAGs $G_s$.

final lower triangular matrix is parametrised as

$$\Phi_{ij} = \begin{cases} \text{MHPA}(\mathbf{R}^{L_2}) \text{ if } i < j, \\ 0 \text{ if } i \geq j, \end{cases} \tag{10}$$

The final decoder architecture that allows for sampling acylic DAGs can be seen in fig. 2.

## 3.3 Loss

Given a permutation matrix $\mathbf{Q}$ and lower triangular binary matrix $\mathbf{A}$ a DAG can be constructed as $\mathcal{G} = \mathbf{Q}\mathbf{A}\mathbf{Q}^T$. As $\mathbf{A}$ is a lower triangular matrix with Bernoulli random variables with parameters $\Phi$, we can write $P_\phi(\mathcal{G}|\mathbf{Q}, \Phi, \mathbf{X}) = \text{Bernoulli}(\mathbf{Q}\Phi\mathbf{Q}^T)$. Since $\mathbf{Q}$ is sampled from the distribution defined in section 3.2, the log probability of a DAG $\mathcal{G}$ is

$$\log P_\phi(\mathcal{G}|\mathbf{X}) = \log \sum_{\mathbf{Q}} P_\phi(\mathcal{G}|\mathbf{Q}, \Phi, \mathbf{X}) P_\phi(\mathbf{Q}|\mathbf{X}) \tag{11}$$

$$\approx \log \frac{1}{S} \sum_{i=1}^{S} P_\phi(\mathcal{G}|\mathbf{Q}_i, \Phi, \mathbf{X}), \quad \mathbf{Q}_i \sim P_\phi(\mathbf{Q}|\mathbf{X}), \tag{12}$$

where $P_\phi(\mathbf{Q}|\mathbf{X}) = \mathcal{GS}(\Theta(\mathbf{X}))$. Combining eq. (3) and eq. (12), the following loss approximates the posterior given datasets and DAGs from the Bayesian causal model:

$$\min_\phi -\mathbb{E}_{P_{\text{BCM}(\mathbf{X})}} \left[ \mathbb{E}_{P_{\text{BCM}(\mathcal{G}|\mathbf{X})}} \left[ \log \frac{1}{S} \sum_{i=1}^{S} P_\phi(\mathcal{G}|\mathbf{Q}_i, \Phi, \mathbf{X}) \right] \right], \quad \mathbf{Q}_i \sim P_\phi(\mathbf{Q}|\mathbf{X}). \tag{13}$$

## 4 Experiments

After defining the *Bayesian causal neural process* (BCNP) model and the loss function in section 3, which approximates the posterior of a Bayesian causal model given samples of datasets and DAGs, we move on to testing the practical performance of the BCNP model. Since real-world datasets with established causal relationships are rare and often disputed (Mooij et al., 2020), we evaluate the BCNP model and baselines using synthetic and semi-synthetic data. With our experiments, we aim to answer the following questions: 1) When the true posterior is known, do the samples from our model match the true posterior? (section 4.1), 2) How does the BCNP model perform in comparison with explicit Bayesian models and other meta-learning models? (section 4.2), 3) How does the BCNP model and other meta-learning models perform on realistic datasets, where the datasets do not directly inform Bayesian causal model and hence the training data? (section 4.3)

**Metrics** We follow prior work in using the *Area Under the Receiver Operating Curve* (AUC) to assess edge prediction at various probability thresholds for the existence of an edge (Friedman & Koller, 2000). Additionally, we evaluate the sum of the *log Bernoulli probabilities* for each edge under the model. The AUC reflects the ranking of edge probabilities, while the log probability metric considers the actual probability values. Both metrics focus on marginal edge probabilities. Previous work also considers metrics like the expected *expected Structural Hamming Distance* (expected SHD) and *expected edge F1 score* over samples. However, we argue that these are not necessarily suitable for evaluating Bayesian methods because they average metrics over the posterior, which doesn't reflect how good a prediction based on the uncertainty *in* the posterior is. Nonetheless, we include them as they offer insights into the expected utility of the samples. Lower is better for expected SHD, while higher is better for the rest.

## 4.1 COMPARING BAYESIAN META-LEARNING METHODS ON TWO VARIABLE NORMALISED LINEAR GAUSSIAN DATA

We consider the two variable $X, Y$ case where datasets are either generated from the DAG $X \rightarrow Y$ or $Y \rightarrow X$. The causes are generated from a normal distribution and the cause to effect relationship is linear with additive Gaussian noise. After generating, both variables are normalised. All the meta-learning models are trained on this data. This is a known unidentifiable case where the data cannot identify the causal graph at all (Peters et al., 2017; Dhir et al., 2024). A sound Bayesian model should thus generate samples of $X \rightarrow Y$ and $Y \rightarrow X$ with probability 0.5 each. To do so will require ensuring that the samples are acyclic, and terms of the adjacency matrix are correlated.

The details of the data generation and model used are in appendix B.1. We generate 100 test datasets and 500 samples from each of the AVICI, CSIvA, and BCNP models. Table 3 shows the proportion of graph types output by each of the models. It shows that while CSIvA and BCNP output reasonable samples, AVICI does not. Although the data is clearly correlated, AVICI outputs samples with no causal relation 32% of the time, and cyclic samples 19% of the time. This is because the decoder in AVICI does not take into account edge dependencies. Even though CSIvA outputs the correct graphs here, it does not explicitly enforce acyclicity or permutation equivariance. In section 4.2, we will see that with more variables its performance suffers.

## 4.2 COMPARISON ON VARIOUS BAYESIAN CAUSAL MODELS

To analyse the performance of the Bayesian meta-learning methods as well as explicit Bayesian models, on larger number of nodes, we compare the models on data from 20 node Bayesian causal models with varying functional distributions. Causal discovery methods tend to perform worse on denser graphs. Therefore, following prior work, we generate graphs from Erdos Renyi (ER) distributions (Erdos et al., 1960) with varying densities, with expected edge counts of 20, 40, and 60. To showcase the advantage of the meta-learning Bayesian approach, we choose functional relationships whose posterior may be hard to approximate with explicit Bayesian models: 1) *Linear:* Linear functions with heteroscedastic noise, 2) *NeuralNet:* Randomly initialised neural networks with a normally distribution latent variable included as an input, 3) *GPCDE:* Random function drawn from a Gaussian process with a latent variable included as an input. The exact details of the data generating procedure is in appendix B.2. For each of the function types and graph densities, we sample 25 datasets to test the methods on.

We compare against the explicit Bayeisan models DiBS (Lorch et al., 2021), BayesDAG (Annadani et al., 2024), as well as the Bayesian meta-learning models AVICI and CSIvA. Each meta-learning model is trained on datasets from each of the functional and graph distributions. Details of all the models are in appendix B.3. Due to memory constraints in the autoregression, the CSIvA model has a lower width than AVICI and BCNP. However we do show that AVICI and BCNP outperform CSIvA with similar widths in appendix C.2. To see if a single model can perform well across multiple distributions, we also include a single BCNP model trained on all synthetic datasets (labelled BCNP All Data).

Results for the densest graphs, ER60, are in table 4. The rest of the results for the Linear datasets are in table 7, table 8, table 9, for the NeuralNet datasets are in table 10, table 11, table 12, and for the GPCDE datasets are in table 10, table 11, table 12. Meta-learning models generally outperform explicit Bayesian models like DiBS and BayesDAG, especially with denser graphs (ER60). Among

meta-learning models, CSIvA underperforms, likely due to the challenges of autoregressive adjacency matrix generation with increased variables, as well as missing permutation equivariance with respect to nodes. The challenge in autoregression is also noted by the authors in Ke et al. (2022), who resort to adding an auxiliary loss. As the AUC and Log Probability only consider marginal edge probabilities, AVICI and BCNP show comparable performance in these metrics. However, as demonstrated in Section section 4.1, AVICI faces difficulties in DAG sampling, an issue not captured by these metrics but partially reflected in the expected edge F1 score. Training on a mixture of all datasets (BCNP All Data) maintains performance across all datasets (more results in appendix C), suggesting that training on a mixture of distributions can help mitigate model misspecification without a loss of performance on each distribution.

## 4.3 COMPARISON ON SYNTREN

Since real data for causal discovery tasks is scarce, we follow prior work and use data generated from simulators. A commonly used simulator is the Syntren generator (Van den Bulcke et al., 2006) which generates gene expression data that matches known experimental data. Each dataset consists of 20 nodes with 500 samples each. We use the 10 datasets generated by Lachapelle et al. (2019) to test all the models. To train the meta-learning models, we use a mixture of all the possible function and graph distributions in section 4.2. Exact details about the training data used and baselines is in appendix B.3.

Table 2 shows that the BCNP is competitive even in the case when the model has not been trained on the exact distribution from which the data is sampled. We found that training on a wide range of distributions helped model performance as well as improving the capacity of the model.

Table 2: Results for the Syntren dataset (mean ± std of mean). Results are over 10 datasets.

| Model | AUC | Log Probability | Expected SHD | Expected Edge F1 |
|---|---|---|---|---|
| DiBS | **0.81 ± 0.02** | -83.31 ± 1.68 | 59.83 ± 1.22 | 0.15 ± 0.01 |
| BayesDAG | 0.52 ± 0.04 | -195.53 ± 16.78 | 99.09 ± 0.90 | 0.10 ± 0.01 |
| AVICI | 0.66 ± 0.03 | -90.82 ± 4.82 | **48.94 ± 2.37** | 0.09 ± 0.01 |
| CSIvA | 0.37 ± 0.03 | -100.59 ± 2.49 | 67.34 ± 1.11 | 0.07 ± 0.00 |
| **BCNP** | **0.84 ± 0.02** | **-76.35 ± 3.61** | **50.87 ± 1.92** | **0.17 ± 0.01** |

## 5 DISCUSSION AND LIMITATIONS

Our objective minimises the KL-divergence between the posterior of the Bayesian causal model from which training data is drawn $P_{\text{BCM}}(\mathcal{G}|\mathbf{X})$, and the BCNP model $P_\phi(\mathcal{G}|\mathbf{X})$. Datasets and their corresponding causal graphs may also come from an unknown distribution $P_{\text{BCM}}(\mathcal{G}|\mathbf{X})$ instead of an explicit causal model. Nevertheless, given a dataset from $P_{\text{BCM}}(\mathcal{G}|\mathbf{X})$, if our model achieves a KL-divergence of zero, we can expect it to perform well on downstream tasks requiring the posterior from $P_{\text{BCM}}(\mathcal{G}|\mathbf{X})$. However, given datasets from a *separate* distribution $\Pi$, what can we say about the performance of our approximate posterior $P_\phi(\mathcal{G}|\mathbf{X})$? Note that

$$\text{KL}[\Pi(\mathcal{G}|\mathbf{X})\|P_{\text{BCM}}(\mathcal{G}|\mathbf{X})] + \text{KL}[P_{\text{BCM}}(\mathcal{G}|\mathbf{X})\|P_\phi(\mathcal{G}|\mathbf{X})] = 0 \implies \text{KL}[\Pi(\mathcal{G}|\mathbf{X})\|P_\phi(\mathcal{G}|\mathbf{X})] = 0.$$

If the KL-divergence between our model and the BCM is zero, then the performance on datasets drawn from $\Pi$ will depend on how closely $P_{\text{BCM}}(\mathcal{G}|\mathbf{X})$ matches the target distribution $\Pi(\mathcal{G}|\mathbf{X})$. If the KL-divergence is not zero, the approximation is likely perform even worse on tasks requiring $\Pi(\mathcal{G}|\mathbf{X})$. Thus, to improve performance, either the BCNP training should be enhanced through more data and higher capacity, or the training data distribution $P_{\text{BCM}}(\mathcal{G}|\mathbf{X})$ should be adjusted to better align with the target distribution. Although results from section 4.3 conform to this intuition a more rigorous treatment of this question would be necessary to make stronger claims.

Unlike other meta-learning approaches (Ke et al., 2022; Lorch et al., 2022), we do not incorporate interventional data in this work. However, similar to these studies, interventional data can be integrated into the BCNP model by appending an indicator vector to the node axis in section 3.1 , signaling whether a node has been intervened upon.

## 6 CONCLUSION

In this work, we address challenges in causal discovery by employing a Bayesian framework to quantify uncertainty. Meta-learning approaches are useful here as they circumvent complex functional inference and enable efficient sampling from high-dimensional spaces. Building on prior research, we incorporate key properties of the posterior distribution such as permutation equivariance with respect to the nodes and dependency between edges. In contrast to other meta-learning methods, the BCNP model also provides samples of DAGs from the posterior. The BCNP model demonstrates competitive performance compared to both explicit Bayesian and other meta-learning models. Our work suggests that meta-learning holds significant potential for a tractable and accurate Bayesian treatment of causal discovery.

Table 3: Proportion of different graphs output in the two-variable unidentifiable case tested with 50% $X_1 \rightarrow X_2$ and 50% $X_2 \rightarrow X_1$. The table shows that as AVICI only estimates the marginal probability of each edge, sampling from it does not output graphs from the correct posterior.

| DAG Type | AVICI | CSIvA | BCNP |
|---|---|---|---|
| No edge between $X_1$ and $X_2$ | 0.3194 | 0.0000 | 0.0124 |
| $X_1 \rightarrow X_2$ | 0.2587 | 0.4851 | 0.4979 |
| $X_2 \rightarrow X_1$ | 0.2342 | 0.5147 | 0.4897 |
| $X_1 \leftrightarrow X_2$ (bidirectional) | 0.1877 | 0.0000 | 0.0000 |
| Other | 0.0000 | 0.0002 | 0.0000 |

Table 4: Results for the 20 variable Linear ER60, NeuralNet ER60, and GPCDE ER60 datasets (mean ± std of mean). The BCNP ER60 model has been trained on datasets from the specific function type and graph density, while the BCNP All Data is trained on a mixture of all datasets.

| Linear ER60 Dataset | | | |
|---|---|---|---|
| **Model** | **AUC** | **Log Probability** | **Expected SHD** | **Expected Edge F1** |
| DiBS | 0.40 ± 0.01 | -231.40 ± 3.69 | 106.64 ± 1.25 | 0.11 ± 0.00 |
| BayesDAG | 0.51 ± 0.01 | -1509.27 ± 48.40 | **63.51 ± 0.40** | 0.04 ± 0.01 |
| AVICI | 0.83 ± 0.01 | -126.31 ± 1.71 | 80.90 ± 0.71 | 0.31 ± 0.01 |
| CSIvA | 0.49 ± 0.01 | -169.49 ± 0.19 | 102.00 ± 0.05 | 0.15 ± 0.00 |
| **BCNP ER60** | **0.85 ± 0.01** | **-121.87 ± 2.29** | 76.52 ± 0.92 | **0.36 ± 0.01** |
| **BCNP All Data** | **0.86 ± 0.01** | **-122.02 ± 2.66** | 72.15 ± 1.17 | **0.36 ± 0.01** |

| NeuralNet ER60 Dataset | | | |
|---|---|---|---|
| **Model** | **AUC** | **Log Probability** | **Expected SHD** | **Expected Edge F1** |
| DiBS | 0.64 ± 0.01 | -208.22 ± 1.82 | **65.72 ± 0.17** | 0.13 ± 0.01 |
| BayesDAG | 0.58 ± 0.01 | -170.73 ± 0.26 | 103.95 ± 0.09 | 0.19 ± 0.01 |
| AVICI | 0.79 ± 0.01 | -138.04 ± 1.36 | 85.15 ± 0.59 | 0.29 ± 0.01 |
| CSIvA | 0.50 ± 0.01 | -169.30 ± 0.20 | 101.89 ± 0.05 | 0.15 ± 0.00 |
| **BCNP ER60** | **0.85 ± 0.00** | **-121.09 ± 1.08** | 70.40 ± 0.35 | **0.40 ± 0.00** |
| **BCNP All Data** | 0.83 ± 0.01 | -127.48 ± 2.71 | 68.34 ± 0.91 | **0.39 ± 0.01** |

| GPCDE ER60 Dataset | | | |
|---|---|---|---|
| **Model** | **AUC** | **Log Probability** | **Expected SHD** | **Expected Edge F1** |
| DiBS | 0.67 ± 0.01 | -199.87 ± 2.03 | **64.22 ± 0.22** | 0.13 ± 0.00 |
| BayesDAG | 0.66 ± 0.01 | -184.48 ± 5.12 | 107.46 ± 0.31 | **0.24 ± 0.00** |
| AVICI | 0.71 ± 0.01 | -151.70 ± 1.28 | 91.33 ± 0.36 | 0.21 ± 0.01 |
| CSIvA | 0.49 ± 0.01 | -169.53 ± 0.19 | 101.62 ± 0.05 | 0.15 ± 0.00 |
| **BCNP ER60** | 0.72 ± 0.01 | -149.48 ± 1.05 | 93.07 ± 0.34 | 0.22 ± 0.01 |
| **BCNP All Data** | **0.77 ± 0.01** | **-142.97 ± 1.80** | 80.28 ± 0.77 | **0.25 ± 0.01** |

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

# A  ADDITIONAL BACKGROUND

## A.1  TRANSFORMERS

Here we describe a transformer layer Vaswani et al. (2017).

Given an input from a previous layer $\mathbf{X} \in \mathbb{R}^{N \times d_{\text{model}}}$, the key $\mathbf{K}$, query $\mathbf{Q}$, and value $\mathbf{V}$ matrices are computed as

$$\mathbf{K} = \mathbf{X}\mathbf{W}^k, \tag{14}$$

$$\mathbf{Q} = \mathbf{X}\mathbf{W}^q, \tag{15}$$

$$\mathbf{V} = \mathbf{X}\mathbf{W}^v, \tag{16}$$

where $\mathbf{W}^k, \mathbf{W}^q, \mathbf{W}^v$ are the key, query, and value weights with $\mathbf{K} \in \mathbb{R}^{N \times h_k}$, $\mathbf{Q} \in \mathbb{R}^{N \times h_k}$, and $\mathbf{V} \in \mathbb{R}^{N \times h_v}$. We define the dot-product attention on these matrices as

$$\text{Attn}(\mathbf{Q}, \mathbf{K}, \mathbf{V}) := \text{softmax}(\mathbf{Q}\mathbf{K}^T / \sqrt{h_k})\mathbf{V}, \tag{17}$$

where the softmax is over the last dimension. Multi-head attention is where multiple query, key, and value pairs are used (indexed by $m$), their results concatenated and the result projected down using $\mathbf{W}^o \in \mathbb{R}^{M \times h_v}$

$$\text{MHSA}(\mathbf{X}) := \text{stack}_{m=1,\dots,M}(\text{head}_m(\mathbf{X}))\mathbf{W}^o, \tag{18}$$

$$\text{where } \text{head}_m(\mathbf{X}) = \text{Attn}(\mathbf{Q}_m, \mathbf{K}_m, \mathbf{V}_m), \tag{19}$$

where $\mathbf{Q}_m = \mathbf{X}\mathbf{W}_m^k, \mathbf{K}_m = \mathbf{X}\mathbf{W}_m^k$, and $\mathbf{V}_m = \mathbf{X}\mathbf{W}_m^v$ The final transformer layer is as follows

$$\text{R}(\mathbf{X}) := \mathbf{X} + \text{MHSA}(\text{LN}(\mathbf{X})), \tag{20}$$

$$\text{T}(\mathbf{X}) := \text{R}(\mathbf{X}) + \text{MLP}(\text{R}(\mathbf{X})), \tag{21}$$

where $\text{LN}(\cdot)$ is a layer-normalisation operation (Ba et al., 2016), and $\text{MLP}$ is a feed-forward layer.

A key property of the attention operation is that permuting the rows of $\mathbf{X}$ results in the rows of the output of the attention being permuted the same way. That is, applying a permutation $\mathbf{P}$ matrix to the input of the layer $\tilde{\mathbf{X}} = \mathbf{PX}$ results in,

$$\tilde{\mathbf{K}} = \tilde{\mathbf{X}}\mathbf{W}^k, \tag{22}$$

$$\tilde{\mathbf{Q}} = \tilde{\mathbf{X}}\mathbf{W}^q, \tag{23}$$

$$\tilde{\mathbf{V}} = \tilde{\mathbf{X}}\mathbf{W}^v, \tag{24}$$

and

$$\tilde{\mathbf{Q}}\tilde{\mathbf{K}}^T = \mathbf{PQK}^T\mathbf{P}^T, \tag{25}$$

$$\text{Softmax}(\tilde{\mathbf{Q}}\tilde{\mathbf{K}}^T)\tilde{\mathbf{V}} = \mathbf{P}\left(\text{Softmax}(\mathbf{QK}^T)\mathbf{V}\right) \tag{26}$$

$$= \mathbf{P}\,\text{Attn}(\mathbf{Q}, \mathbf{K}, \mathbf{V}). \tag{27}$$

The transformer architecture thus encodes permutation equivariance with respect rows of the inputs Lee et al. (2019).

## B  EXPERIMENT DETAILS

### B.1  TWO VARIABLE LINEAR GAUSSIAN EXPERIMENTS DETAILS

We generate the data by first sampling a graph for two variables $X, Y$ that either corresponds to $X \to Y$ or $Y \to X$. For an example graph of $X \to Y$, we generate the cause and effects as

$$X \sim \mathcal{N}(0, 1), \tag{28}$$

$$w \sim \mathcal{N}(0, 10), \tag{29}$$

$$\sigma \sim \text{Gamma}(2.5, 2.5), \tag{30}$$

$$Y \sim \mathcal{N}(wX, \sigma^2). \tag{31}$$

After generation, we normalise both $X$ and $Y$ to ensure that the causal direction is not identifiable from data. We generate $200,000$ datasets in total with $1,000$ samples each.

For all the models, we use Adam (Kingma & Ba, 2014) with a learning rate of $10^{-4}$ and a batch size of $64$. We train for $2$ epochs and use a linear warmup for $10\%$ of the total iterations. To reduce the memory footprint, we train in bfloat $16$. All models used $4$ layers in their encoder, while BCNP and CSIvA used $4$ decoder layers. AVICI directly decodes from the summary representation. For BCNP, we used $100$ permutation samples to approximate the loss, and maximum of $1000$ sinkhorn iterations.

### B.2  BAYESIAN CAUSAL MODEL DATA GENERATION DETAILS

Here, we provide details about the data generation process used to generate training and test data for results in section 4.2. Following recommendations from Reisach et al. (2021), we normalise all variables after generation.

All datasets contain 1000 samples with 20 nodes. For training the Bayesian meta-learning models, we generate $500,000$ datasets. Each test set contains 25 datasets. We generate graphs from the ER distribution Erdos et al. (1960), and generate datasets for varying densities $20, 40, 60$.

***Linear*:**  Given a graph, we sample all causes from $\mathcal{N}(0, 1)$. Samples for each variable $X_d$ are then generated as follows

$$w \sim \mathcal{N}(0, 10), \tag{32}$$

$$\sigma_i \sim \text{Gamma}(2.5, 2.5), \tag{33}$$

$$(X_d)_i = \mathcal{N}\left(w^T\left(X_{\text{PA}_{\mathcal{G}}(d)}\right)_i, \sigma_i\right), \tag{34}$$

where $\left(X_{\text{PA}_{\mathcal{G}}(d)}\right)_i$ is the $i^{th}$ sample of the parents of variable $X_d$ in graph $\mathcal{G}$.

*NeuralNet*: We sample each variable as follows

$$\epsilon_i \sim \mathcal{N}(0, 1), \tag{35}$$

$$(X_d)_i = \text{NN}_\theta \left( \left( X_{\text{PA}_\mathcal{G}(d)} \right)_i, \epsilon_i \right), \tag{36}$$

where $\left( X_{\text{PA}_\mathcal{G}(d)} \right)_i$ is the $i^{th}$ sample of the parents of variable $X_d$ in graph $\mathcal{G}$ and $\text{NN}_\theta$ is a randomly initialised neural network. We use a two layer network with Leaky Relu activation and a width of 32. When a node has no parents, we simply pass the noise $\epsilon$ through the neural network to generate the variable.

*GPCDE*: Given samples of parents of a node $X_{\text{PA}_\mathcal{G}(d)}$ and a scalar noise term $\epsilon \sim \mathcal{N}(0, 1)$, we first sample hyperparameters

$$\gamma \sim \text{Uniform}(0.1, 1), \tag{37}$$

$$\boldsymbol{\lambda} \sim \log \mathcal{N}(-1, 1), \tag{38}$$

where one $\lambda$ is generated per input (including the scalar noise term), $\boldsymbol{\lambda} = \left[ \lambda_1, \ldots, \lambda_{|X_{\text{PA}_\mathcal{G}(d)}|+1} \right]$. The exponential Gamma kernel is then constructed (Rasmussen, 2003)

$$K \left( \left( \left( X_{\text{PA}_\mathcal{G}(d)} \right)_i, \epsilon_i \right), \left( \left( X_{\text{PA}_\mathcal{G}(d)} \right)_j, \epsilon_j \right) \right) = \tag{39}$$

$$\exp \left( -\frac{1}{\boldsymbol{\lambda}} \left\| \left( \left( X_{\text{PA}_\mathcal{G}(d)} \right)_i, \epsilon_i \right) - \left( \left( X_{\text{PA}_\mathcal{G}(d)} \right)_j, \epsilon_j \right) \right\|_2^{2\gamma} \right) \tag{40}$$

where one $\lambda$ is generated per input. Sampling the $\gamma$ parameters ensures that we sample functions with varying smoothness. Sampling a different $\lambda$ for each input variable changes how quickly the function changes with respect to an input. Both together ensure that we sample a large variety of functions. We then sample the variable as follows

$$\sigma \sim \text{Gamma}(1, 10), \tag{41}$$

$$X_d \sim \mathcal{N} \left( 0, K + \sigma^2 \mathbb{I} \right). \tag{42}$$

### B.3 BAYESIAN CAUSAL MODEL AND REAL DATA EXPERIMENT DETAILS

Here, we provide the hyperparameter settings and baseline details for the experiments carried out in section 4.2. We compare against the explicit Bayesian models BayesDAG (Annadani et al., 2024) and DiBS (Lorch et al., 2021). Here, the authors provide different linear and non-linear estimators. The non-linear estimators are neural networks. BayesDAG performs inference over the neural network weights by using SG-MCMC (Ma et al., 2015), whereas DiBS using Stein variational gradient descent (Liu & Wang, 2016). We also compare against the Bayesian meta-learning models AVICI (Lorch et al., 2022) and CSIvA (Ke et al., 2022). The differences between our model BCNP and the other Bayesian meta-learning models can be seen in table 1.

**DiBS:** [1]

Table 5: Hyperparameters for the DiBS model for the Linear, NeuralNet, GPCDE, and Syntren datasets

| Hyperparameters | Linear | NeuralNet | GPCDE | Syntren |
|---|---|---|---|---|
| $\alpha$ | 0.2 | 0.02 | 0.02 | 0.2 |
| $\gamma_z$ | 5 | 5 | 5 | 5 |
| $\gamma_\theta$ | 500 | 1000 | 1000 | 500 |
| Number of particles | 64 | 32 | 32 | 32 |

DiBS provides a linear and a non-linear model. The hyperparameters include the annealing rate of the acyclicity regulariser term $\alpha$, lengthscale of $\gamma_z$ for the latent kernel, and a lengthscale of $\gamma_\theta$ for

---

[1] https://github.com/larslorch/dibs

the parameter kernel For both models, we use the tuned hyperparameters from Lorch et al. (2021) for 20 variables for the Bayesian causal model datasets. For the Syntren dataset, we use values tuned in Annadani et al. (2024). All the hyperparameters used can be seen in table 5. We provide all the models with the correct graph density, number of variables as well as the graph generating distribution (ER) to construct its prior. For Syntren we used the scale-free (Barabási & Albert, 1999) graph prior, as it closely matches the structure of the Syntren datasets. Due to the high memory cost, we optimised 64 particles for the linear model and 32 for the non-linear model. As the number of samples are quite low, when these were averaged to find the probability of each edge, the probability of certain edges was 0. This lead to an infinite log Bernoulli probability value. To deal with this, during testing, we add constants to the log probability from $10^{-8}$ to $10^{-2}$ and pick the highest resultant log probability. We keep all other hyperparameters to the preset value by the authors.

**BayesDAG:** [2] For the Syntren dataset, we used the values tuned by the authors. For the Bayesian causal model datasets, following Annadani et al. (2024), for each function type, we generate 5 validation sets with graphs drawn from the ER distribution (Erdos et al., 1960) with 20 expected edges. For the linear and the non-linear models, and for each validation set we sample 20 hyperparameter configurations from the recommended hyperparameter range settings in the authors' the code. In contrast to the settings used by authors, we normalise the all variables as recommended by Reisach et al. (2021). We note that the authors recommend choosing hyperparameters based on the lowest expected SHD score. However, in our experience this lead to nearly empty graphs on the test set. Hence, we choose hyperparameters based on the highest AUC score. The chosen hyperparameters for each function type can be seen in table 6. We keep all other hyperparameters to their preset values.

Table 6: Hyperparameters for the BayesDAG model for the Linear, NeuralNet, GPCDE, and Syntren datasets

| Hyperparameters | Linear | NeuralNet | GPCDE | Syntren |
|---|---|---|---|---|
| $\lambda_s$ | 500.0 | 10.0 | 1.0 | 300.0 |
| Number of Chains | 20 | 10 | 10 | 10 |
| Scale $\Theta$ | 0.001 | 0.001 | 0.1 | 0.01 |
| Scale $\mathbf{p}$ | 1.0 | 0.01 | 0.001 | 0.1 |

**Bayesian Meta-Learning Models:** For the models AVICI, CSIvA, and ours BCNP, we ensure the same architecture here we highlight key differences. AVICI uses max pooling to construct the summary representation, instead of attention using a query vector. The summary representation is decoded using a linear layer (Lorch et al., 2022). AVICI also uses an acyclic regulariser on it's marginal edge probabilities which forces the probabilities to be acyclic. Following the values used by the authors, we initialise the weight of the acyclic regulariser to 0 and update it every 500 iterations using a regulariser learning rate (Lorch et al., 2022). This regulariser learning rate is kept at a value of $10^{-4}$ after warming up for $10\%$ of the total iterations. In contrast, the only difference with CSIvA is that it uses an autoregressive decoder where each $D^2$ elements of the adjacency are autoregressively generated. All the models have 2 encoder layers (4 transformer layers in total, but attention over samples and nodes is done twice each), while CSIvA, and BCNP also have 4 decoder layers. For AVICI and BCNP, we use a width of 512 for the attention layers and a width of 1024 for the feedforward layers. Due to memory constraints of autoregressive generation, for CSIvA we use a width of 256 for the attention and 512 for the feedforward layers. We use 8 attention heads for each model. We use Adam (Kingma & Ba, 2014) with a learning rate of $10^{-4}$ with a linear warmup of $10\%$ of the total iterations. We use a batch size of 32 for AVICI and BCNP, and a batch size of 8 for CSIvA. For BCNP, we used 100 permutation samples to approximate the loss, and maximum of 1000 sinkhorn iterations.

**Training details for Bayesian meta-learning models for Syntren:** To train the meta-learning models, we generate data from a mixture of all possible choices used in appendix B.2. We generate

---

[2]`https://github.com/microsoft/Project-BayesDAG`

graphs from the ER distribution as well as the scale-free distribution (Barabási & Albert, 1999). The densities of the graphs are sampled from 20 to 60. Each edge function is sampled from the NeuralNet or the GPCDE (described in appendix B.2 distributions with equal probability. As the Syntren datasets have 500 samples per dataset, we sample the same amount. We use 500, 000 datasets to train in total. We use the same networks as for the rest of the experiments, but increase the number of attention heads to 16 and increase the feedforward width of the AVICI and BCNP model to 2048. For CSIvA we kept the same width as the other experiments due to memory constraints.

## C  ADDITIONAL RESULTS

### C.1  RESULTS ON DATA FROM BAYESIAN CAUSAL MODELS

In this section we present the full results for each of the function type and graph density (section 4.2). Higher is better for AUC, Log Probability, and Expected Edge F1, while lower is better for Expected SHD.

For the BCNP model, those labelled ER20, ER40, ER60 are trained on datasets from the same distribution as the test dataset. Those labelled ER20-60 are trained on a mixture of the graph densities for a particular function type (Linear, NeuralNet, GPCDE), hence there is one model per function type. The model labelled All Data, is a single model that is trained on a mixture of all graph densities and function types. The fact that the single model performs as well (sometimes better, e.g. GPCDE datasets), than training on a single distribution type, shows that training on a mixture of datasets can be a useful strategy.

Table 7: Results for the 20 variable Linear ER20 dataset (mean ± std of mean).

| Model | AUC | Log Probability | Expected SHD | Expected Edge F1 |
|---|---|---|---|---|
| DiBs | 0.73 ± 0.01 | -70.53 ± 1.64 | 25.41 ± 0.74 | 0.16 ± 0.01 |
| BayesDAG | 0.52 ± 0.01 | -515.58 ± 18.76 | 23.02 ± 0.54 | 0.03 ± 0.01 |
| AVICI | **0.89 ± 0.01** | **-51.27 ± 1.47** | 26.61 ± 0.59 | 0.31 ± 0.01 |
| CSIvA | 0.51 ± 0.01 | -79.79 ± 0.13 | 38.36 ± 0.01 | 0.05 ± 0.00 |
| **BCNP ER20** | **0.90 ± 0.01** | **-52.11 ± 2.77** | **21.53 ± 0.75** | **0.44 ± 0.02** |
| **BCNP ER20-60** | **0.89 ± 0.01** | **-52.47 ± 2.42** | 23.47 ± 0.74 | **0.41 ± 0.02** |
| **BCNP All Data** | **0.89 ± 0.01** | **-52.07 ± 2.90** | 23.35 ± 0.75 | **0.44 ± 0.02** |

Table 8: Results for the 20 variable Linear ER40 dataset (mean ± std of mean).

| Model | AUC | Log Probability | Expected SHD | Expected Edge F1 |
|---|---|---|---|---|
| DiBS | 0.53 ± 0.02 | -160.71 ± 4.13 | 72.43 ± 2.08 | 0.11 ± 0.01 |
| BayesDAG | 0.51 ± 0.01 | -1016.56 ± 47.11 | **44.07 ± 0.53** | 0.03 ± 0.01 |
| AVICI | 0.86 ± 0.01 | -94.12 ± 1.82 | 56.22 ± 0.48 | 0.28 ± 0.01 |
| CSIvA | 0.51 ± 0.01 | -130.16 ± 0.20 | 71.24 ± 0.03 | 0.10 ± 0.00 |
| **BCNP ER40** | **0.89 ± 0.01** | **-89.38 ± 2.39** | 49.68 ± 0.56 | 0.37 ± 0.01 |
| **BCNP ER20-60** | **0.88 ± 0.01** | **-90.63 ± 2.36** | 51.17 ± 1.04 | 0.36 ± 0.01 |
| **BCNP All Data** | **0.90 ± 0.01** | **-86.49 ± 3.03** | 49.01 ± 1.05 | **0.40 ± 0.01** |

Table 9: Results for the 20 variable Linear ER60 dataset (mean $\pm$ std of mean).

| Model | AUC | Neg. Log Prob. | Expected SHD | Expected Edge F1 |
|---|---|---|---|---|
| DiBS | $0.40 \pm 0.01$ | $-231.40 \pm 3.69$ | $106.64 \pm 1.25$ | $0.11 \pm 0.00$ |
| BayesDAG | $0.51 \pm 0.01$ | $-1509.27 \pm 48.40$ | $\mathbf{63.51 \pm 0.40}$ | $0.04 \pm 0.01$ |
| AVICI | $0.83 \pm 0.01$ | $-126.31 \pm 1.71$ | $80.90 \pm 0.71$ | $0.31 \pm 0.01$ |
| CSIvA | $0.49 \pm 0.01$ | $-169.49 \pm 0.19$ | $102.00 \pm 0.05$ | $0.15 \pm 0.00$ |
| **BCNP ER60** | $\mathbf{0.85 \pm 0.01}$ | $\mathbf{-121.87 \pm 2.29}$ | $76.52 \pm 0.92$ | $\mathbf{0.36 \pm 0.01}$ |
| **BCNP ER20-60** | $\mathbf{0.84 \pm 0.01}$ | $-125.42 \pm 2.17$ | $74.19 \pm 1.08$ | $0.33 \pm 0.01$ |
| **BCNP All Data** | $\mathbf{0.86 \pm 0.01}$ | $\mathbf{-122.02 \pm 2.66}$ | $72.15 \pm 1.17$ | $\mathbf{0.36 \pm 0.01}$ |

Table 10: Results for the 20 variable NeuralNet ER20 dataset (mean $\pm$ std of mean).

| Model | AUC | Log Probability | Expected SHD | Expected Edge F1 |
|---|---|---|---|---|
| DiBS | $0.69 \pm 0.01$ | $-75.31 \pm 1.19$ | $28.48 \pm 0.20$ | $0.12 \pm 0.01$ |
| BayesDAG | $0.59 \pm 0.02$ | $-119.99 \pm 0.27$ | $90.21 \pm 0.20$ | $0.09 \pm 0.00$ |
| AVICI | $0.84 \pm 0.01$ | $-59.21 \pm 1.15$ | $29.51 \pm 0.37$ | $0.24 \pm 0.01$ |
| CSIvA | $0.51 \pm 0.01$ | $-79.82 \pm 0.21$ | $38.01 \pm 0.02$ | $0.05 \pm 0.00$ |
| **BCNP ER20** | $\mathbf{0.88 \pm 0.00}$ | $-56.07 \pm 0.76$ | $\mathbf{26.73 \pm 0.16}$ | $0.26 \pm 0.00$ |
| **BCNP ER20-60** | $\mathbf{0.88 \pm 0.00}$ | $\mathbf{-53.92 \pm 0.98}$ | $27.55 \pm 0.36$ | $\mathbf{0.38 \pm 0.01}$ |
| **BCNP All Data** | $\mathbf{0.88 \pm 0.01}$ | $\mathbf{-54.00 \pm 1.94}$ | $30.05 \pm 0.65$ | $\mathbf{0.36 \pm 0.01}$ |

Table 11: Results for the 20 variable NeuralNet ER40 dataset (mean $\pm$ std of mean).

| Model | AUC | Log Probability | Expected SHD | Expected Edge F1 |
|---|---|---|---|---|
| DiBS | $0.67 \pm 0.01$ | $-139.47 \pm 1.96$ | $\mathbf{46.81 \pm 0.21}$ | $0.13 \pm 0.01$ |
| BayesDAG | $0.59 \pm 0.01$ | $-145.35 \pm 0.18$ | $97.25 \pm 0.14$ | $0.15 \pm 0.00$ |
| AVICI | $0.79 \pm 0.01$ | $-105.83 \pm 1.21$ | $61.11 \pm 0.46$ | $0.24 \pm 0.01$ |
| CSIvA | $0.50 \pm 0.01$ | $-130.44 \pm 0.22$ | $71.92 \pm 0.04$ | $0.10 \pm 0.00$ |
| **BCNP ER40** | $\mathbf{0.86 \pm 0.00}$ | $\mathbf{-93.32 \pm 1.43}$ | $47.38 \pm 0.28$ | $\mathbf{0.39 \pm 0.00}$ |
| **BCNP ER20-60** | $\mathbf{0.86 \pm 0.00}$ | $-95.39 \pm 1.31$ | $49.71 \pm 0.38$ | $0.37 \pm 0.00$ |
| **BCNP All Data** | $0.84 \pm 0.01$ | $-101.49 \pm 3.37$ | $53.44 \pm 0.71$ | $0.37 \pm 0.01$ |

Table 12: Results for the 20 variable NeuralNet ER60 dataset (mean $\pm$ std of mean).

| Model | AUC | Log Probability | Expected SHD | Expected Edge F1 |
|---|---|---|---|---|
| DiBS | $0.64 \pm 0.01$ | $-208.22 \pm 1.82$ | $\mathbf{65.72 \pm 0.17}$ | $0.13 \pm 0.01$ |
| BayesDAG | $0.58 \pm 0.01$ | $-170.73 \pm 0.26$ | $103.95 \pm 0.09$ | $0.19 \pm 0.01$ |
| AVICI | $0.79 \pm 0.01$ | $-138.04 \pm 1.36$ | $85.15 \pm 0.59$ | $0.29 \pm 0.01$ |
| CSIvA | $0.50 \pm 0.01$ | $-169.30 \pm 0.20$ | $101.89 \pm 0.05$ | $0.15 \pm 0.00$ |
| **BCNP ER60** | $\mathbf{0.85 \pm 0.00}$ | $\mathbf{-121.09 \pm 1.08}$ | $70.40 \pm 0.35$ | $\mathbf{0.40 \pm 0.00}$ |
| **BCNP ER20-60** | $0.83 \pm 0.00$ | $-133.36 \pm 1.59$ | $68.34 \pm 0.36$ | $0.38 \pm 0.00$ |
| **BCNP All Data** | $0.83 \pm 0.01$ | $-127.48 \pm 2.71$ | $68.34 \pm 0.91$ | $\mathbf{0.39 \pm 0.01}$ |

Table 13: Results for the 20 variable GPCDE ER20 dataset (mean ± std of mean).

| Model | AUC | Log Probability | Expected SHD | Expected Edge F1 |
|---|---|---|---|---|
| DiBS | $0.80 \pm 0.01$ | **-63.32 $\pm$ 1.24** | **26.33 $\pm$ 0.17** | $0.18 \pm 0.01$ |
| BayesDAG | $0.67 \pm 0.02$ | $-137.93 \pm 2.63$ | $99.69 \pm 0.22$ | $0.11 \pm 0.00$ |
| AVICI | $0.74 \pm 0.01$ | $-71.67 \pm 0.64$ | $36.27 \pm 0.20$ | $0.08 \pm 0.01$ |
| CSIvA | $0.48 \pm 0.01$ | $-80.19 \pm 0.13$ | $37.93 \pm 0.01$ | $0.05 \pm 0.00$ |
| **BCNP ER20** | $0.75 \pm 0.01$ | $-72.29 \pm 0.87$ | $37.31 \pm 0.10$ | $0.09 \pm 0.00$ |
| **BCNP ER20-60** | $0.74 \pm 0.01$ | $-73.29 \pm 0.92$ | $41.23 \pm 1.01$ | $0.09 \pm 0.00$ |
| **BCNP All Data** | **0.83 $\pm$ 0.01** | **-63.99 $\pm$ 1.42** | $38.24 \pm 0.85$ | **0.20 $\pm$ 0.01** |

Table 14: Results for the 20 variable GPCDE ER40 dataset (mean ± std of mean).

| Model | AUC | Log Probability | Expected SHD | Expected Edge F1 |
|---|---|---|---|---|
| DiBS | $0.74 \pm 0.01$ | $-125.15 \pm 1.52$ | **45.04 $\pm$ 0.26** | $0.16 \pm 0.00$ |
| BayesDAG | $0.67 \pm 0.02$ | $-154.09 \pm 2.25$ | $103.09 \pm 0.33$ | $0.19 \pm 0.01$ |
| AVICI | $0.72 \pm 0.01$ | $-116.67 \pm 1.21$ | $67.68 \pm 0.33$ | $0.15 \pm 0.01$ |
| CSIvA | $0.51 \pm 0.01$ | $-130.29 \pm 0.22$ | $71.65 \pm 0.04$ | $0.10 \pm 0.00$ |
| **BCNP ER40** | $0.73 \pm 0.01$ | $-116.11 \pm 1.26$ | $69.68 \pm 0.30$ | $0.16 \pm 0.00$ |
| **BCNP ER20-60** | $0.73 \pm 0.01$ | $-116.91 \pm 1.33$ | $66.95 \pm 1.10$ | $0.15 \pm 0.00$ |
| **BCNP All Data** | **0.78 $\pm$ 0.01** | **-108.50 $\pm$ 1.58** | $62.05 \pm 1.17$ | **0.22 $\pm$ 0.01** |

Table 15: Results for the 20 variable GPCDE ER60 dataset (mean ± std of mean).

| Model | AUC | Log Probability | Expected SHD | Expected Edge F1 |
|---|---|---|---|---|
| DiBS | $0.67 \pm 0.01$ | $-199.87 \pm 2.03$ | **64.22 $\pm$ 0.22** | $0.13 \pm 0.00$ |
| BayesDAG | $0.66 \pm 0.01$ | $-184.48 \pm 5.12$ | $107.46 \pm 0.31$ | **0.24 $\pm$ 0.00** |
| AVICI | $0.71 \pm 0.01$ | $-151.70 \pm 1.28$ | $91.33 \pm 0.36$ | $0.21 \pm 0.01$ |
| CSIvA | $0.49 \pm 0.01$ | $-169.53 \pm 0.19$ | $101.62 \pm 0.05$ | $0.15 \pm 0.00$ |
| **BCNP ER60** | $0.72 \pm 0.01$ | $-149.48 \pm 1.05$ | $93.07 \pm 0.34$ | $0.22 \pm 0.01$ |
| **BCNP ER20-60** | $0.71 \pm 0.01$ | $-153.97 \pm 1.07$ | $87.03 \pm 0.67$ | $0.19 \pm 0.01$ |
| **BCNP All Data** | **0.77 $\pm$ 0.01** | **-142.97 $\pm$ 1.80** | $80.28 \pm 0.77$ | **0.25 $\pm$ 0.01** |

## C.2 RESULTS FOR META-LEARNING MODELS WITH THE SAME WIDTH

Due to memory constraints, we used a smaller width network for CSIvA than AVICI and BCNP. Here, we train the same width networks for all three to allow for fair comparison. The results show that CSIvA struggles with larger number of nodes.

Table 16: Results for the same width meta-learning models for Linear ER20 (mean ± std of mean).

| Model | AUC | Log Probability | Expected SHD | Expected Edge F1 |
|---|---|---|---|---|
| CSIvA | 0.51 ± 0.01 | -79.79 ± 0.13 | 38.36 ± 0.01 | 0.05 ± 0.00 |
| AVICI | 0.85 ± 0.01 | -61.28 ± 1.54 | 31.91 ± 0.33 | 0.18 ± 0.01 |
| **BCNP** | **0.87 ± 0.01** | **-54.52 ± 1.56** | **26.68 ± 0.62** | **0.31 ± 0.02** |

Table 17: Results for the same width meta-learning models for Linear ER40 (mean ± std of mean).

| Model | AUC | Log Probability | Expected SHD | Expected Edge F1 |
|---|---|---|---|---|
| CSIvA | 0.51 ± 0.01 | -130.16 ± 0.20 | 71.24 ± 0.03 | 0.10 ± 0.00 |
| AVICI | 0.82 ± 0.01 | -102.57 ± 0.96 | 60.25 ± 0.34 | 0.21 ± 0.01 |
| **BCNP** | **0.85 ± 0.01** | **-98.64 ± 1.70** | **57.44 ± 0.45** | **0.27 ± 0.01** |

Table 18: Results for the same width meta learning models for Linear ER60 (mean ± variance of mean).

| Model | AUC | Log Probability | Expected SHD | Expected Edge F1 |
|---|---|---|---|---|
| CSIvA | 0.49 ± 0.00 | -169.49 ± 0.03 | 102.00 ± 0.00 | 0.15 ± 0.00 |
| AVICI | 0.80 ± 0.00 | -134.23 ± 1.87 | 84.83 ± 0.25 | 0.27 ± 0.00 |
| **BCNP** | **0.82 ± 0.00** | **-130.28 ± 5.09** | **82.19 ± 0.30** | **0.31 ± 0.00** |

Table 19: Results for the same width meta-learning models for NeuralNet ER20 (mean ± std of mean).

| Model | AUC | Log Probability | Expected SHD | Expected Edge F1 |
|---|---|---|---|---|
| CSIvA | 0.51 ± 0.01 | -79.82 ± 0.21 | 38.01 ± 0.02 | 0.05 ± 0.00 |
| AVICI | **0.78 ± 0.01** | **-68.97 ± 1.13** | **33.96 ± 0.30** | **0.11 ± 0.00** |
| **BCNP** | 0.70 ± 0.01 | -73.86 ± 0.65 | 37.46 ± 0.19 | 0.08 ± 0.00 |

Table 20: Results for the same width meta-learning models for NeuralNet ER40 (mean ± std of mean).

| Model | AUC | Log Probability | Expected SHD | Expected Edge F1 |
|---|---|---|---|---|
| CSIvA | 0.50 ± 0.01 | -130.44 ± 0.22 | 71.92 ± 0.04 | 0.10 ± 0.00 |
| AVICI | **0.73 ± 0.01** | **-115.88 ± 0.88** | 64.97 ± 0.37 | **0.16 ± 0.00** |
| **BCNP** | Model diverged | Model diverged | Model diverged | Model diverged |

Table 21: Results for the same width meta-learning models for NeuralNet ER60 (mean ± std of mean).

| Model | AUC | Log Probability | Expected SHD | Expected Edge F1 |
|---|---|---|---|---|
| CSIvA | 0.50 ± 0.01 | -169.30 ± 0.20 | 101.89 ± 0.05 | 0.15 ± 0.00 |
| AVICI | 0.75 ± 0.01 | -146.07 ± 1.25 | 88.62 ± 0.48 | 0.23 ± 0.00 |
| **BCNP** | **0.77 ± 0.01** | **-141.54 ± 1.51** | **87.53 ± 0.49** | **0.26 ± 0.00** |

Table 22: Results for the same width meta learning models for GPCDE ER20 (mean ± std of mean).

| Model | AUC | Log Probability | Expected SHD | Expected Edge F1 |
|---|---|---|---|---|
| CSIvA | 0.48 ± 0.01 | -80.19 ± 0.13 | 37.93 ± 0.01 | 0.05 ± 0.00 |
| AVICI | **0.74 ± 0.01** | **-72.94 ± 0.64** | 36.26 ± 0.20 | **0.08 ± 0.00** |
| **BCNP** | **0.74 ± 0.01** | **-72.85 ± 0.87** | **35.38 ± 0.10** | **0.08 ± 0.00** |

Table 23: Results for the same width meta learning models for GPCDE ER40 (mean ± std of mean).

| Model | AUC | Log Probability | Expected SHD | Expected Edge F1 |
|---|---|---|---|---|
| CSIvA | 0.51 ± 0.01 | -130.29 ± 0.22 | 71.65 ± 0.04 | 0.10 ± 0.00 |
| **AVICI** | **0.72 ± 0.01** | **-116.70 ± 1.21** | **67.11 ± 0.33** | **0.15 ± 0.00** |
| **BCNP** | **0.72 ± 0.01** | **-116.72 ± 1.22** | **67.00 ± 0.30** | **0.15 ± 0.00** |

Table 24: Results for the same width meta learning models for GPCDE ER60 (mean ± std of mean).

| Model | AUC | Log Probability | Expected SHD | Expected Edge F1 |
|---|---|---|---|---|
| CSIvA | 0.49 ± 0.01 | -169.53 ± 0.19 | 101.62 ± 0.05 | 0.15 ± 0.00 |
| **AVICI** | **0.71 ± 0.01** | **-151.76 ± 1.28** | **92.79 ± 0.36** | **0.21 ± 0.00** |
| **BCNP** | 0.70 ± 0.01 | **-152.24 ± 0.76** | 94.27 ± 0.34 | **0.21 ± 0.00** |

