# OpenReview forum: "A Meta-Learning Approach to Bayesian Causal Discovery"
_ICLR.cc/2025/Conference — ICLR 2025 Poster_

### Official Review · Reviewer_DiiJ · 2024-11-01

**Soundness:** 3
**Presentation:** 3
**Contribution:** 2
**Rating:** 6
**Confidence:** 4

**Summary:**

The paper introduces the Bayesian Causal Neural Process (BCNP), a novel approach for Bayesian causal discovery that leverages meta-learning to address challenges in traditional methods. BCNP learns a mapping from datasets to posterior distributions over causal graphs, thereby bypassing the need to explicitly infer the posterior over causal mechanisms and allowing for efficient sampling from high-dimensional spaces. By encoding key properties like permutation equivariance and edge dependencies, BCNP accurately approximates the true posterior and generates acyclic graph samples. The paper demonstrates BCNP's effectiveness through experiments on synthetic and semi-synthetic datasets, showcasing its superior performance compared to existing Bayesian and meta-learning models for causal discovery.

**Strengths:**

* The paper is well-written and clear and includes a range of well-designed experiments.

**Weaknesses:**

For me, there are two major concerns regarding the novelty and contributions of the proposed method to the community:

1. The paper presents the direct mapping from data to a posterior over graphs ($P(G|D)$) as a main advantage, bypassing the need to model causal mechanisms. However, learning a distribution over causal mechanisms is crucial for many applications, such as causal inference and treatment effect estimation [1, 2]. Could the authors elaborate on potential applications of the proposed approach beyond the discovery of the causal graph?
2. The proposed method builds on several modules from existing work. For instance, the representation of the posterior distribution as permutation and upper triangular matrices is also similar to [3]. Although the method outperforms baselines in graph-based metrics in causal discovery, the specific novelty remains unclear, particularly given the first concern.

[1] Emezue, C. C., et al. (2023). Benchmarking Bayesian Causal Discovery Methods for Downstream Treatment Effect Estimation. arXiv, 2307.04988. https://arxiv.org/abs/2307.04988v3

[2] Schölkopf, B., et al. (2021). Towards Causal Representation Learning. arXiv, 2102.11107. https://arxiv.org/abs/2102.11107v1

[3] Cundy, C., et al. (2021). BCD Nets: Scalable Variational Approaches for Bayesian Causal Discovery. arXiv, 2112.02761. https://arxiv.org/abs/2112.02761v1

**Questions:**

1. As my main concerns, I would appreciate the authors to elaborate on the abovementioned weaknesses.
2. Given that the proposed method does not utilize interventional data, using CPDAG E-SHD (as used in previous papers such as in BayesDAG) might be more reflective of the performance of the models compared with E-SHD.
3. Have the authors tried any experiments on low data regime where the number of samples is close to the number of nodes? This is particularly important since Bayesian Causal Discovery is most desirable in low data regime where $n \approx d$. Do you have an intuition about how the proposed model performs in these particular settings?
4. In several experiments, other baselines such as BayesDAG and DiBS achieve a better E-SHD compared to the proposed method. Could you elaborate on why this happens?

---

> ### Author Response · Authors · 2024-11-21
> **Response to Reviewer DiiJ (1)**
>
> We thank the reviewer for their comments that "The paper is well-written and clear and includes a range of well-designed experiments."
> We address your questions point by point below.
>
>
> >  Learning a distribution over causal mechanisms is crucial for many applications, such as causal inference and treatment effect estimation. Could the authors elaborate on potential applications of the proposed approach beyond the discovery of the causal graph?
>
> Estimating causal effects is an important downstream application of causal discovery.
> In our case, causal effects can be estimated by **sampling DAGs and then performing do-calculus using a chosen estimator**.
> In fact, this is exactly the procedure to infer causal effects for all the methods in the paper you have cited [1] (Figure 1 and section 3.2 in [1]).
> Plenty of the methods tested in **[1] also only infer the causal structure** - Bootstrap-PC and DAG-GFlowNet to name a few.
> Furthermore, a lot of other (non-Bayesian) causal discovery methods also only infer the causal graph [2, 3, 4].
>
>
> [1] Emezue, C. C., et al. (2023). Benchmarking Bayesian Causal Discovery Methods for Downstream Treatment Effect Estimation.
> [2] Rolland, Paul, et al. "Score matching enables causal discovery of nonlinear additive noise models." International Conference on Machine Learning. PMLR, 2022.
> [3] Spirtes, Peter. "An anytime algorithm for causal inference." International Workshop on Artificial Intelligence and Statistics. PMLR, 2001.
> [4] Bühlmann, Peter, Jonas Peters, and Jan Ernest. "CAM: Causal additive models, high-dimensional order search and penalized regression." The Annals of Statistics 42.6 (2014).
>
> >  For instance, the representation of the posterior distribution as permutation and upper triangular matrices is also similar to [3]
>
> We invite the reviewer to refer to the general comment for our summary of our main contribution.
>
> It is true that plenty of methods use the permutation and upper triangular construction (discussed in L133), however the **difference is the method used for inference**.
> Whereas previous methods use sampling or variational inference with explicit Bayesian models, we use meta-learning to infer the posterior.
>
> In contrast with other meta-learning approaches (which only parametrised a single graph), we target the full Bayesian posterior and hence include important properties (in Table 1).
> The permutation and upper triangular construction provides us with these properties so was included in the decoder construction (see L264) along with a novel loss to infer the posterior using meta-learning (section 3.3).
>
> An important contribution of our work is that, **with our changes, meta-learning approaches can better approximate the posterior over causal structure than explicit Bayesian models**.
> This can especially be seen in Table 2, where with the same training data as our model (BCNP), previous approaches AVICI and CSIvA do not outperform the explicit Bayesian models (DiBS and BayesDAG), whereas our model (BCNP) does.
>
>
> >...using CPDAG E-SHD (as used in previous papers such as in BayesDAG) might be more reflective of the performance
>
> The CPDAG is useful when the underlying causal model is known to be *not* identifiable (which is the case in sec. 6.1 of BayesDAG which is the only section where CPDAG E-SHD is used).
> The causal models that we generate data from have been previously used for testing discovery *within* a Markov equivalence class (functional mechanisms are similar to those used in [1, 2]).
> Hence, we believe that E-SHD is more appropriate.
>
> [1] Goudet, Olivier, et al. "Learning functional causal models with generative neural networks." Explainable and interpretable models in computer vision and machine learning (2018).
>
> [2] Dhir, Anish, Samuel Power, and Mark van der Wilk. "Bivariate Causal Discovery using Bayesian Model Selection." Forty-first International Conference on Machine Learning.
>
>
> >  Do you have an intuition about how the proposed model performs in these [low data] particular settings?
>
> The sample size of the training datasets can easily be changed to better reflect low data settings (see Appendix B.2).
> The performance of meta-learning models depends on the _number of datasets_ used for training.
> As the training data is synthetically generated, **any number of training datasets can be generated**.
> Thus, we would still expect performance gains over other explicit Bayesian models.

---

> > ### Author Response · Authors · 2024-11-21
> > **Response to Reviewer DiiJ (2)**
> >
> > > other baselines such as BayesDAG and DiBS achieve a better E-SHD compared to the proposed method
> >
> > This is an interesting point which explains why the **SHD alone is probably not a good metric for causal discovery**.
> > The SHD of an empty graph with respect to a graph with density 60 will be 60.
> > DiBS and BayesDAG sometimes result in much sparser graphs, which leads to low SHD scores (around the value of the density of the graph) (see L828).
> > This is also evidenced by very low E-F1 scores of these methods, for example in Table 4, which suggests that they are not identifying correct edges.
> >
> >
> > We hope we have answered your questions and addressed your concerns.
> > We would be happy to discuss any of the points further.

---

> > > ### Comment · Reviewer_DiiJ · 2024-11-25
> > >
> > > Thank you for the responses. Considering the other reviews and the authors' responses, I now lean towards accepting the paper and will raise my score accordingly.

---

### Official Review · Reviewer_fWq7 · 2024-11-04

**Soundness:** 3
**Presentation:** 2
**Contribution:** 2
**Rating:** 6
**Confidence:** 2

**Summary:**

In Bayesian causal discovery, the goal is to approximate the posterior over graphs. Since computing the posterior over graphs requires computing posterior over functional parameters, previous approaches have tried to approximate the posterior over function parameters.  This paper uses neural processes to approximate the posterior over graphs directly by learning a map from the dataset to the set of distributions. The neural process is an encoder-decoder network that claims to sample, permutation-invariant, acyclic graphs that capture edge dependencies.

**Strengths:**

The paper proposes a technique that can directly sample from the posterior over graphs. The main contribution of the paper is in the architecture of the encoder-decoder network that captures necessary properties like the permutation invariance by cross-attention, and ayclic DAGs and edge dependencies by sampling the decomposition of a DAG into permutation and lower-triangular matrices. The comparison of the proposed model is done with existing meta-learning approaches and Bayesian approaches using multiple metrics. The proposed model is shown to be competitive on synthetic and simulated datasets. The paper is written clearly and flows well.

**Weaknesses:**

My main concern is that of this being an incremental contribution. The framework is not new and the only contributions I see are a change in how permutation invariance and acyclicity is incorporated. Please let me know if I am missing something else. I have a detailed list of questions below whose answers might help strengthen the paper.

**Questions:**

1) The ideas that the paper uses to impose acyclicity and permutation invariance seem applicable in general. Can existing meta-learning approaches be modified to get the same? While this is not the focus of the paper. It would make it stronger to highlight it if that's the case.

2) The experimental validation for the case when the true posterior is known, is done only for the two-variable case. Is it possible to scale this up?

3) How can the posterior over the graphs be used for inference on that of the functional parameters?

4) How is the performance on Gene Regulatory Network simulated data like SERGIO? I believe it would be useful to add that in given its importance.

5) The paper says that the metrics average over the posterior. I believe this is a problem with any metric. Are there alternatives that the authors propose?

---

> ### Author Response · Authors · 2024-11-21
> **Response to Reviewer fWq7**
>
> We thank the reviewer for their comments that "The proposed model is shown to be competitive on synthetic and simulated datasets" and "The paper is written clearly and flows well".
> We answer your questions below.
>
> > The framework is not new and the only contributions I see are a change in how permutation invariance and acyclicity is incorporated.
>
> Actually, we disagree with the reviewer on this point. We are actually the **first method to use meta-learning to directly target an approximation of the full Bayesian posterior and yield accurate samples from this approximate posterior**.
>
> Causal structures can only be represented by acyclic graphs.
> Methods like CSIvA and AVICI do not guarantee acyclic graph samples.
> CSIvA also does not encode permutation equivariance.
> Further, AVICI only targets the marginal probabilities so outputs samples from the wrong posterior in very simple cases (section 4.1).
>
> Please see our general response for more details on this point.
>
>
>
> > The ideas that the paper uses to impose acyclicity and permutation invariance seem applicable in general. Can existing meta-learning approaches be modified to get the same?
>
> This was the exactly starting point for our work.
> We wondered about this too before we decided it was necessary to develop an alternative method.
>
> AVICI encodes permutation equivariance with respect to node while the autoregressive decoder architecture of CSIvA prohibits direct encoding of permutation equivariance.
>
> In general, **there is no clear way to ensure that we can sample from the correct posterior and obtain acyclic samples for AVICI or CSIvA**.
> As AVICI was only used to output a single graph, an acyclicity regulariser is used directly on the edge probabilities, which may lead to an acyclic final graph if the probabilities are thresholded (set probability of over 0.5 to 1).
> However, using this regulariser also biases the posterior probability estimates.
> In the example of sec. 4.1, as the two graphs are completely unidentifiable, the true posterior should have marginal probabilities $P(A_{12}) = 0.5$ and $P(A_{21}) = 0.5$.
> Directly using an acyclicity regulariser on the probabilities promotes one of the terms to go higher and the other to go lower, leading to an _inaccurate posterior_.
> The same applies to the output probabilities of CSIvA.
>
>
> >  The experimental validation for the case when the true posterior is known, is done only for the two-variable case. Is it possible to scale this up?
>
> The point of this experiment was to show the clear downsides of the other meta-learning approaches (speciically AVICI).
> This is best visualised with the two variable linear Gaussian case where the true posterior is obvious.
>
> In general, the true posterior can be found for multiple variable cases, but only for the linear Gaussian model with a specific prior [1].
>
> [1] Geiger, Dan, and David Heckerman. "Parameter priors for directed acyclic graphical models and the characterization of several probability distributions." The Annals of Statistics 30.5 (2002).
>
> > How can the posterior over the graphs be used for inference on that of the functional parameters?
>
> We're not sure if the posterior over the causal graph can be used **directly** to infer the functional parameters.
>
> The functional parameters for a specific causal structure can be found by sampling a causal structure and using a different estimator to infer the functional parameters.
>
> For downstream tasks such as causal effect estimation, [1] provides an example of how the posterior over causal structures can be used.
>
> [1] Emezue, C. C., et al. (2023). Benchmarking Bayesian Causal Discovery Methods for Downstream Treatment Effect Estimation.
>
> > Gene Regulatory Network simulated data like SERGIO
>
> Thank you for this suggestion.
> Syntren is exactly a gene expression network generated from a simulator.
> Is this sufficient or is SERGIO completely different?
>
> > The paper says that the metrics average over the posterior. I believe this is a problem with any metric. Are there alternatives that the authors propose?
>
> This is an interesting quesiton.
> The first two metrics AUC and Log probability do not average over the posterior, however they only take into account the marginal edge probabilities.
> We believe there is scope for further research into properly evaluating Bayesian causal discovery methods.
>
> We hope we have answered the reveiwers questions.
> We're happy to discuss any point further or answer any other questions.

---

> > ### Comment · Reviewer_fWq7 · 2024-11-26
> >
> > Sorry for my delayed follow-up. Thank you for your detailed response.
> >
> >  "Actually, we disagree with the reviewer on this point. We are actually the first method to use meta-learning to directly target an approximation of the full Bayesian posterior and yield accurate samples from this approximate posterior."
> >
> > Line 155-156 say, "Methods such as AVICI and CSIvA aim to approximate the posterior over the causal structure (Ke
> > et al., 2022; Lorch et al., 2022)." The first part of your claim in the rebuttal seems to contradict this statement in the paper? The second part is what I also consider as a contribution of yours.
> >
> > I am still not convinced enough about the significance to increase my score.

---

> > > ### Author Response · Authors · 2024-11-26
> > > **Response to Reviewer fWq7**
> > >
> > > >The first part of your claim in the rebuttal seems to contradict this statement in the paper? The second part is what I also consider as a contribution of yours.
> > >
> > > We apologise for the confusion and agree that our original statement in the paper is not clear. AVICI and CSIvA **say** that they are targeting the posterior over causal structures but they only target the single maximum a posteriori (MAP) graph, and **not the full posterior**.
> > > We will adjust that statement in the paper to be "Methods such as AVICI and CSIvA aim to approximate the posterior over the causal structure but only target the maximum a posteriori (MAP) graph and not the full posterior".
> > >
> > > >I am still not convinced enough about the significance to increase my score.
> > >
> > > We believe our work is significant as it is the first meta-learning model to target the **full posterior** (not just the MAP value) and follow through on the implications of targeting the full posterior.
> > > The advantage of this can be seen in the better performance over other meta-learning models.
> > >
> > > Methods like AVICI (Ke et al., 2022) and CSIvA (Lorch et al., 2022) are mainly tested in interventional causal discovery settings, where interventions provide enough information to completely identify the causal graph, hence the full posterior is not needed.
> > > In contrast, our work focuses on approximating the full posterior distribution over causal structures in observational causal discovery scenarios. In these settings, identifiability issues prevent the unique determination of the causal graph from observational data alone, making it **essential to consider the entire posterior distribution rather than just the MAP estimate**.
> > >
> > > Attempting to use AVICI and CSIvA for Bayesian tasks where the full posterior estimation is required, shows that they lack key properties necessary for a practical Bayesian causal discovery method (see table 1). For instance, evaluating expectations for downstream tasks necessitates sampling valid **acyclic graphs** from the posterior distribution, which are necessary for representing valid causal structures. However, both **AVICI and CSIvA fail to consistently provide acyclic samples**.
> > > Moreover, we demonstrate that sampling from the correct posterior is not achievable with AVICI even in very simple cases (section 4.1). CSIvA also struggles in performance when dealing with a higher number of variables (shown in results tables), limiting its scalability and effectiveness in more complex scenarios.
> > >
> > > Our approach addresses these shortcomings by accurately approximating the **full posterior instead of just the MAP value** and enabling reliable sampling of valid acyclic graphs, thereby fulfilling the essential criteria for a usable Bayesian causal discovery method.
> > > This, coupled with the significant performance improvements over current state-of-the art methods justify our contribution.

---

> > > > ### Author Response · Authors · 2024-12-02
> > > > **We kindly ask if the rebuttal has addressed the reviewer's concerns**
> > > >
> > > > Thank you for engaging with our work.
> > > >
> > > > We wanted to enquire whether our clarification of the statement and the proposed changes had addressed your concerns?
> > > >
> > > > We are happy to discuss any point further if not.

---

### Official Review · Reviewer_CqB9 · 2024-11-06

**Soundness:** 3
**Presentation:** 4
**Contribution:** 3
**Rating:** 6
**Confidence:** 4

**Summary:**

This paper introduces a Bayesian meta-learning model for causal discovery, named the Bayesian Causal Neural Process. This "meta-learning" algorithm learns a model from pairs of datasets and their associated causal graphs. Specifically, the model incorporates a neural process to address model uncertainty and employs an encoder-decoder architecture carefully designed to achieve desirable properties for causal discovery in a meta-learning context. These properties include permutation invariance to sample order, permutation equivariance to nodes, and acyclicity, achieved by decomposing the adjacency matrix into a lower triangular matrix and a permutation matrix—some of which build on existing approaches. The authors demonstrate the empirical superiority of this model over other explicit Bayesian and Bayesian meta-learning models

**Strengths:**

- As summarized in Table 1, this approach implements several key desiderata for a Bayesian meta-learning model more effectively than existing alternatives.
- The paper is well-written, allowing readers to easily follow the motivation and desiderata of Bayesian meta-learning models.

**Weaknesses:**

- While I appreciate the overall architecture of the model, it is challenging to identify which components provide truly novel technical contributions. For instance, the results on pages 4 and 5 appear to largely rely on existing findings, and, if I'm not mistaken, page 6 includes results from Annadani et al. (2024). The contributions of this paper are not clearly or explicitly articulated.
- The Bayesian prior is learned through pairs of datasets and directed acyclic graphs (DAGs). From my perspective, learning P(X,G) is likely very challenging. Indeed, Appendix B.1 notes that, for a case involving two variables, the model was trained on 200,000 datasets. I understand that Section 4.1 is intended to evaluate the posterior, but in Appendix B.2, 500,000 datasets are used for training. This information would be better suited for the main text, and details on computational resources should be transparently reported there as well (they are not provided even in the appendix).
- I find it difficult to be fully convinced by the meta-learning approach to causal discovery. While I understand the importance of establishing a strong Bayesian prior, can we realistically expect access to hundreds of thousands of dataset-graph pairs?

**Questions:**

- For DiBS and BayesDAG, the authors mention that they “keep all other hyperparameters to their preset values.” What if these hyperparameters were tuned? Would the results differ significantly? For instance, DiBS and BayesDAG perform well on the Syntren dataset for certain metrics, so tuning may further widen this performance gap.
- In the caption, it says, “Each dataset contains D nodes and N samples.” Does this imply that all datasets must be of the same size? What if the sample sizes vary, such as 100, 200, or 500? Should N be set to 100, or can larger datasets (e.g., 200 and 500 samples) be split into multiple parts (e.g., 2 or 5 parts) to train the model? In that case, should there be weighting adjustments (1/2, 1/5), also training perhaps only for data-related parameters and not for those related to the graph prior?
- If there is no training involved, does your model function like a standard Bayesian causal discovery method with a uniform prior?
- Each variable might have distinct functional priors, such as some being linear, others non-linear, discrete, or having different means. Does the encoder “implicitly” reorder these variables (perhaps using attention mechanisms) to align them by their functional type? I’m trying to conceptually grasp how the encoder handles such diverse functional characteristics under the hood.
- What if permutation equivariance among nodes was not enforced? Would this potentially improve performance, especially if nodes are ordered consistently between training and test data?
- In Line 382, could you clarify what is meant by “correlations between edges” when dealing with just two variables, X and Y, with at most a single edge?

Minor comments
- It would be better to have some annotations in Figure 2 (given that the figure takes lots of space)
- Not quite sure whether it is sufficient to say the dependence in permutation and lower triangular binary matrices is modeled because they share the same representation R0.
- 047, 137 citep for Wenzel et al.
- 115 citing PC, GES might be good (given you have some space for 119)
- 141 Bayesian metal → meta
- 305 Is the grammar correct? (The matrix … and thus … )
- 340 space before Mena
- 389 In section section 4.2
- 389 its
- 414 space after comma
- 480 enable
- 485 period
- BayesDAG is not highlighted in Table 4 for Linear data/SHD
- Figure 2 shouldn’t o be the input to Theta? or onto the edge between R^L1 to Theta?
- Sorry for suggesting my preference here but can we align Theta and Phi (parameters, vertically) and align Qs and As with the same horizontal line as Theta and Phi, respectively, and put Gs at far right.

(I can change to weak accept (or accept) depending on the response and others' reviews.)

---

> ### Author Response · Authors · 2024-11-21
> **Response to Reviewer CqB9 (1)**
>
> We thank the reviewer for their comment that our method "implements several key desiderata for a Bayesian meta-learning model more effectively than existing alternatives" and our paper is "well-written, allowing readers to easily follow the motivation".
> We address your questions below.
>
> >  it is challenging to identify which components provide truly novel technical contributions
>
>
> We invite the reviewer to refer to the general comment for our summary of our main contribution.
>
> Whereas previous meta-learning approaches were only concerned with a single graph, we parametrise the full posterior over causal structures.
> This lead us to include key properties of the posterior that the other methods missed (Table 1, section 4.1).
> The key changes compared to the architecture of the previous approaches include the **decoder (L264) to include the above properties and the loss function (section 3.3) to train the network using meta-learning**.
>
> Our change to the decoder to include the key properties of the posterior resulted in using the permutation lower triangular construction.
> Whereas many methods have used this construction (see L133), they use variational inference or sampling based methods with explicit Bayesian models to infer the posterior.
>
> The promise of meta-learning is to approximate the posterior more accurately than explicit Bayesian models (see L103, L199).
> We show that previous methods (AVICI, CSIvA) fall short of this promise.
> With the same training data as our model (BCNP), previous approaches such as AVICI and CSIvA do not outperform explicit Bayesian models DiBS and BayesDAG (e.g. Table 2).
> **With our changes, our model BCNP outperforms explicit Bayesian models**.
>
>
> > From my perspective, learning P(X,G) is likely very challenging. Indeed, Appendix B.1 notes that, for a case involving two variables, the model was trained on 200,000 datasets. I understand that Section 4.1 is intended to evaluate the posterior, but in Appendix B.2, 500,000 datasets are used for training. This information would be better suited for the main text, and details on computational resources should be transparently reported there as well (they are not provided even in the appendix).
>
> We used an A100 GPU to train these models. We will include this in Section 4.
> Note that the data used for training these models is **entirely synthetic** (see Appendix B.2), hence as many datasets as required can be generated.
> Once a model is trained, inference for a new dataset is simply a forward pass.
> We agree that learning $P(X,G)$ is challenging, however our results show that the meta-learning approach, if done properly, is a viable path for Bayesian causal discovery.
>
> If you have any more specific questions regarding this, or any comments that you think will help improve the paper, we would appreciate it.
>
> > can we realistically expect access to hundreds of thousands of dataset-graph pairs?
>
> All the experiments were carried out by training on synthetic data which was generated from a pre-defined causal model (and hence any amount can be generated, Appendix B.2).
>
> Worth noting is that the results in table 2 (on syntren) **did not use training data from the syntren generator, but from the causal models defined in Appendix B.2** (see L861 for exact details).
> This shows that training on synthetic data and testing on entirely different distributions still gives good performance.
>
> Known causal relations from real data, which will be much less in number, may be incorporated into an already trained model to learn an even better prior, but we leave how to finetune these models for future work.
>
> Do you have any other specific questions that might help assuage any concerns about the meta-learning approach?
>
>
> > For DiBS and BayesDAG, the authors mention that they “keep all other hyperparameters to their preset values.” What if these hyperparameters were tuned? Would the results differ significantly?
>
> Note that **we tune all the hyperparameters that the authors tune for their work** (or use tuned values from the authors where relevant).
> In fact we tune more hyperparameters for these baselines than we do for our model, where we only tune the learning rate.
> We also tune the learning rate **only once for our model but tune the hyperparameters once for each dataset type for the baselines**.
> Hence we do not believe the performance difference can be attributed to hyperparameter tuning.
>
> The preset hyperparameters are things like the architecture (residual connections, normalisation layers etc.) and training steps that we set to the value used by the authors in their work.
>
>
>
> > Does this imply that all datasets must be of the same size? What if the sample sizes vary, such as 100, 200, or 500?
>
> This is an interesting question.
> The datasets do not have to be the same size.
> Variable node and sample sizes can easily be handled by padding approppriately.
> This is commonly done for attention models when processing variable length sequences.

---

> > ### Author Response · Authors · 2024-11-21
> > **Response to Reviewer CqB9 (2)**
> >
> > > If there is no training involved, does your model function like a standard Bayesian causal discovery method with a uniform prior?
> >
> > Could you please elaborate on this question? We're a bit unsure what you mean here.
> >
> >
> > > I’m trying to conceptually grasp how the encoder handles such diverse functional characteristics under the hood.
> >
> > This is an interesting question.
> > If multiple datasets with different functional types are used, it may be that the summary representation $R^0$ of similar functional types are clustered together.
> > Although there are other aspects that the encoder might cluster more on, for example the graph structure itself.
> >
> >
> > > What if permutation equivariance among nodes was not enforced? Would this potentially improve performance, especially if nodes are ordered consistently between training and test data?
> >
> >
> > Again, this is an interesting question and an important consideration.
> > We do not think performance will improve without permutation equivariance. Permutation equivariance is a key property as the ordering of the input to the model should not affect the posterior.
> > For example, consider a model trained on two variable synthetic data and then tested on measurements of altitude (A) and temperature (T). Whether we input (A, T) or (T, A) should not change the belief over causal structures.
> > There is no "consistent ordering" in this case.
> >
> >
> > > In Line 382, could you clarify what is meant by “correlations between edges” when dealing with just two variables, X and Y, with at most a single edge?
> >
> > Thank you for pointing this out, this should say "correlation between terms of the adjacency matrix".
> > We will change this.
> > The true posterior in Section 4.1 is correlated as if $A_{12} = 1$, this implies that $A_{21} = 0$ and vice versa (where $A$ is the adjacency matrix).
> > AVICI only models the marginal edge probabilities so samples $A_{12}$ indpendently of $A_{21}$, leading to the samples from the wrong distribution in Table 3.
> >
> >
> > > Minor comments
> >
> > Thank you for pointing these out, we will make these changes.
> >
> > We hope we have satisfactorily answered your questions.
> > We're happy to discuss any of the points further.

---

> > ### Comment · Reviewer_CqB9 · 2024-11-25
> >
> > Thanks for clear answers. I will adjust my score based on the general response and response to my review.

---

### Official Review · Reviewer_xBXp · 2024-11-08

**Soundness:** 4
**Presentation:** 3
**Contribution:** 4
**Rating:** 6
**Confidence:** 3

**Summary:**

This paper proposes the Bayesian Causal Neural Process (BCNP), a Neural Process-based framework for learning a posterior distribution over causal graphs given a dataset $P(G \mid X)$ . The proposed method is scalable and captures distinctions between graphs within the same Markov equivalence class. Since BCNP learns causal structures across multiple (synthetic) datasets, it is considered a meta-learning approach.

**Strengths:**

1. This paper provides a well-structured and self-contained summary of prior works in Bayesian causal discovery, allowing readers to clearly follow the evolution of Bayesian approaches for learning causal graphs. As a reviewer, this summary enables me to track the advancements in the field and understand the recent progress in addressing challenges like scalability and uncertainty in causal inference.

2. The paper clearly outlines its unique contributions in Table 1, highlighting significant advancements over existing methods. The proposed model addresses critical aspects such as acyclicity, permutation invariance, and edge dependencies—all of which are critical for accurate causal graph learning. These improvements over prior approaches directly enhance the reliability and precision of inferred causal structures, making the contributions both clear and impactful.

3. The proposed BCNP is particularly attractive due to its innovative use of Neural Processes to directly sample the posterior distribution $ P(G | X) $. This approach enables BCNP to achieve scalability and uncertainty-aware inference by learning across tasks (= datasets), allowing the model to efficiently adapt to new datasets while maintaining a robust representation of causal uncertainty. This combination of scalability, direct posterior sampling, and adaptability positions BCNP as a strong advancement over existing Bayesian causal discovery methods.

**Weaknesses:**

1. One of the main limitations of this paper is the assumption that there are no latent confounders between variables. This assumption may limit the applicability of the method to real-world datasets where unobserved confounders are common.

2. I think this method is heavily relying on the assumptions on the choice of modeling class of functions F, the types of graphs and the noise model. Unlike fully nonparametric approaches like the PC algorithm, which do not impose strict functional or noise assumptions, BCNP’s effectiveness depends on these choices being well-suited to the data.

3. The proposed framework may be sensitive to hyperparameter settings and architecture choices, such as the hyperparameters for the encoder-decoder network, or the types of prior distributions. Fine-tuning these parameters is often data-dependent, which could reduce the model's robustness and generalization capabiltiy.

**Questions:**

1. Is it nontrivial to consider the setting where latent confounders between variables exist?

2. Is the proposed method computationally efficient when a function class is chosen as neural networks?

3. How sensitive the proposed framework is regarding to the sparcity of the graph?

4. Are there any interesting example where the Bayesian approach beats the Markov-equivalence-based method?

---

> ### Author Response · Authors · 2024-11-21
> **Response to Reviewer xBXp**
>
> We thank the reviewer for their comments that "The paper clearly outlines its unique contributions" and that our paper "positions BCNP as a strong advancement over existing Bayesian causal discovery methods".
> We address your questions below.
>
> >One of the main limitations of this paper is the assumption that there are no latent confounders between variables. This assumption may limit the applicability of the method to real-world datasets where unobserved confounders are common.
> > Is it nontrivial to consider the setting where latent confounders between variables exist?
>
> Actually, it is very nontrivial to consider latent confounders [Ch. 9, 1].
> When and why we can identify graphs with latent confounders is currently an open problem.
> The few methods that claim to identify latent confounders make very strong assumptions when doing so.
>
> [1] Peters, Jonas, Dominik Janzing, and Bernhard Schölkopf. Elements of causal inference: foundations and learning algorithms. The MIT Press, 2017.
>
>
> > Unlike fully nonparametric approaches like the PC algorithm, which do not impose strict functional or noise assumptions, BCNP’s effectiveness depends on these choices being well-suited to the data.
>
> Methods such as PC are limited as they only use conditional independence information and hence can **only identify a causal structure upto a Markov equivalence class** [Ch. 4, 1].
> Bayesian methods can distinguish **within a Markov equivalence class** [2].
>
> The effectiveness of BCNP, like any Bayesian method, depends on how good the prior is for the data, and how well the posterior is approximated (Section 5).
> The advantage here is that **we can use any functional mechanisms with priors that allow for sampling**, where the posterior might be hard to approximate.
>
> When there is no knowledge of the data generating process, a wider prior should be considered (Section 5).
> This is the approach we follow in Section 4.3.
> Here, the training data is **not informed by the syntren generator and we still outperform all other methods** (details in B.3).
> Thus, section 4.3 shows that our model can perform well even without knowledge of the true data distribution.
>
>
> [1] Peters, Jonas, Dominik Janzing, and Bernhard Schölkopf. Elements of causal inference: foundations and learning algorithms. The MIT Press, 2017.
>
> [2] Dhir, Anish, Samuel Power, and Mark van der Wilk. "Bivariate Causal Discovery using Bayesian Model Selection." Forty-first International Conference on Machine Learning.
>
> > The proposed framework may be sensitive to hyperparameter settings and architecture choices, such as the hyperparameters for the encoder-decoder network, or the types of prior distributions. Fine-tuning these parameters is often data-dependent, which could reduce the model's robustness and generalization capabiltiy.
>
>
> For the meta-learning models, we only tune the learning rate **once** by looking at the loss on a synthetically generated validation set.
> The same hyperparameters was used for all experiments.
> Hence the hyperparameters used were not dataset dependent.
> This is contrast with the baseline methods (DiBS and BayesDAG) where several hyperparameters were tuned for each dataset type (details in B.3).
>
> The performance of our method *does* depend on the types of causal models from which training datasets are sampled (as they form the prior).
> However, as we show in Section 4.3, when we have no knowledge of the data generating distribution, sampling training data from a mixture of causal models (representing a wider prior) performs well (L861 has exact details).
>
>
> > Is the proposed method computationally efficient when a function class is chosen as neural networks?
>
> A nice feature of meta-learning is that
> **the computational performance is independent of the function class** that is used to generate the training data.
>
>
> > How sensitive the proposed framework is regarding to the sparcity of the graph?
>
> We test this in our experiments.
> See C.1 for results on varying density of graphs.
> Our model performs better than other methods as the density increases.
>
> > Are there any interesting example where the Bayesian approach beats the Markov-equivalence-based method?
>
>
> [Ch.4, 1] relates how causal discovery can be done within a Markov equivalence class and [2] relates to how the Bayesian approach is able to do it.
> In general, methods that can identify within a Markov equivalence class tend to do better than Markov-equivalence-based methods.
> This is because methods like PC cannot differentiate graphs like $X \to Y$ and $Y \to X$.
>
>
> [1] Peters, Jonas, Dominik Janzing, and Bernhard Schölkopf. Elements of causal inference: foundations and learning algorithms. The MIT Press, 2017.
>
> [2] Dhir, Anish, Samuel Power, and Mark van der Wilk. "Bivariate Causal Discovery using Bayesian Model Selection." Forty-first International Conference on Machine Learning.
>
>
> We hope we have answered your questions and would be happy to discuss any point further.

---

> > ### Author Response · Authors · 2024-11-29
> > **We kindly ask if the rebuttal has addressed the reviewer's concerns**
> >
> > We appreciate the feedback provided. As the discussion period is coming to a close, we wanted to enquire if our rebuttal had addressed your concerns.
> >
> > We are also happy to discuss any point further.

---

### Author Response · Authors · 2024-11-21
**Common Response**

We thank the reviewers for taking the time to review our paper.
We appreciate that they consider the contributions "both clear and impactful" `#xBXp`, that the paper is well-written `#DiiJ,#fWq7,#CqB9`, and that our proposed model is competitive `#fWq7`.
We also appreciate comments that the experiments were well-designed `#DiiJ`, that our approach "implements several key desiderata" `#CqB9`, and that our work is a "strong advancement over existing Bayesian causal discovery methods" `#xBXp`.


### Contribution

We propose the first model that combines meta-learning and a fully Bayesian approach to causal discovery to parametrise the full Bayesian posterior over causal structures.
Our method is the **only one that provides not only accurate samples, but also samples from the correctly parametrised space** (acyclic graphs with depdendent edges) with permutation equivariance over nodes baked-in (Table 1 and Section 4.1).
We thus show that following the consequences of a fully Bayesian view of meta-learning prescribes changes that actually leads to better performance (results in Section 4). We believe this is an important novelty.

The _promise_ of meta-learning is that it can remove the accuracy bottleneck of approximating an explicit posterior. So if meta-learning is working correctly, it should perform better than existing explicit Bayesian methods.
The existing methods that have attempted meta-learning approaches before (AVICI, CSIvA) were only concerned with parametrizing a single graph rather than the full distribution.
This lead to methodological choices (see L155, Table 1) that mean that **these methods do not parametrise a distribution over the correct object (acyclic graphs with dependent edges) and result in innacurate approximations of the Bayesian posterior**.
This is shown in Section 4.1, and is evident by the fact that these methods (AVICI, CSIvA) **do not outperform explicit Bayesian models** (e.g. Table 2).

In contrast to the above, our method targets an approximation of the full Bayesian posterior by **proposing a decoder (L264)** that captures important properties of the posterior (L155, Table 1) and we provide a **novel loss** to train the network using meta-learning (section 3.3). We then show that this provides samples from the correctly parametrised space (acyclic graphs with dependent edges) and **outperforms explicit Bayesian models** (Table 2).
Our changes, prescribed by the fully Bayesian view, are thus important to **deliver on the promises of meta-learning**.

---

### Author Response · Authors · 2024-11-29
**Revised Manuscript Uploaded**

We thank all the reviewers for their constructive feedback that have helped improve this paper.

We have uploaded a revised manuscript, with changes marked in red, that hopefully address all the reviewers concerns.

Should any further questions or concerns arise, we are happy to provide clarification during the remaining discussion time.

---

### Meta-Review · Area_Chair_hBRB · 2024-12-21

**Metareview:**

The authors propose a heuristic algorithm to approximately learn the posterior distribution over causal graphs given observational data, by posing the problem as a meta-learning objective. Compared to learning the ML or MAP estimate DAG, this objective is much harder but I agree can carry value for certain downstream tasks. So a useful heuristic is valuable, which seems to have been demonstrated via experiments and convinced the reviewers at least to some degree.

Based on my reading, I would say please populate Bayesian causal discovery literature. There is a very large body of work for Bayesian causal discovery but the current citations are very, very limited. Citing any related work is just as important as any other part of paper-writing since not citing properly breaks the connection the next generation of researchers we hope will have to the past.

**Additional Comments On Reviewer Discussion:**

All reviewers are leaning towards acceptance after the rebuttal and no important complaints are left.

---

### Decision · Program_Chairs · 2025-01-22

Accept (Poster)